# Mouse germline cysts contain a fusome-like structure that mediates oocyte development

Madhulika Pathak, Allan C Spradling*

Howard Hughes Medical Institute Research Laboratories, Carnegie Institution for Science, Baltimore, United States

## eLife Assessment

This manuscript provides evidence that mouse germline cysts develop an asymmetric Golgi, ER, and microtubule-associated structure that resembles the fusome in Drosophila germline cysts. This **fundamental** study provides new evidence that fusome-like structures exist in germ cell cysts across species. Overall, the data are **convincing** and represent a significant advance in our understanding of germ cell biology.

*For correspondence:
spradling@carnegiescience.edu

Competing interest: The authors declare that no competing interests exist.

**Abstract** Mouse female primordial germ cells (PGCs) undergo five synchronous, incomplete mitotic divisions and send each resulting germline cyst into meiosis to fragment and produce 4–6 oocytes and 24–26 supportive nurse cells. However, no system of polarity has been found to specify mammalian oocytes, link them appropriately to nurse cells and enable them to acquire high-quality organelles and cytoplasm. We report that mouse cysts develop an asymmetric Golgi, endoplasmic reticulum (ER), and microtubule-associated 'fusome,' similar to the oocyte-determining fusome in *Drosophila* cysts. The mouse fusome distributes asymmetrically among cyst cells and enriches in future oocytes with Pard3 and Golgi-endosomal UPR (unfolded protein response) proteins. Spindle remnants rich in stable acetylated microtubules, like those building the *Drosophila* and Xenopus fusomes, transiently link early mouse cyst cells for part of each cell cycle. A non-random gap in these microtubules predicts that initial cysts fragment into similar six-cell derivatives, providing a potential mechanism for producing uniform oocytes. Together with previous studies, these results argue that a polarized fusome underlies the development of female gametes from the PGC to follicular oocyte stages in diverse animals including mammals.

## Introduction

Early female germ cell development occurs within interconnected cysts formed by consecutive incomplete divisions of a germline stem cell (GSC) or primordial germ cell (PGC) in diverse animals. These include basal species like hydra (*Littlefield, 1994*), nematodes like *C. elegans*, insects like *Drosophila*, and vertebrates including fish, frogs, mice, and humans (*Gondos, 1973*; *King, 1970*; *Lin et al., 1994*; *Pepling and Spradling, 1998*; *Kloc et al., 2004a*; *Marlow and Mullins, 2008*; review, *Spradling, 2024*). As they develop, *Drosophila* and other insect cyst cells become interconnected by ring canals and acquire polarity associated with a microtubule-rich 'fusome' (*Giardina, 1901*; *King, 1970*; *Lin et al., 1994*; *Büning, 1994*; *Huynh and St Johnston, 2004*). In *Drosophila*, the fusome arises from mitotic spindle remnants generated during incomplete cytokinesis, contains ER cisternae and displays Par gene and microtubule-dependent polarity that is essential for oocyte production. Later, in meiotic prophase, the fusome mediates the movement of centrioles, mitochondria, and germ plasm

regulators, such as *oskar* mRNA into the oocyte where they form the Balbiani body (Bb) (*Mahowald and Strassheim, 1970*; *Cox and Spradling, 2003*). Remarkably, mammalian oocytes undergo these same events (review: *Spradling et al., 2022*), but a fusome has not been detected, raising the possibility that mammalian cyst polarity and oocyte specification use a different system.

Studies in Xenopus long provided suggestions that cyst polarization in at least some vertebrate species resembles the invertebrate process (*Kloc et al., 2004a*; review: *Huynh and St Johnston, 2004*). Recently, a detailed study using modern methods characterized fusome-containing Xenopus cysts sharing many such similarities, including production of relatively few oocytes and mostly nurse cells (*Davidian and Spradling, 2025*). Here, we report new studies on mouse cyst polarization. In the mouse, germ cells are induced around embryonic day 6.5 (E6.5) (*Lawson et al., 1999*) and begin migrating toward the gonad as PGCs. En route, PGCs start to reprogram their epigenetic state to pluripotency (*Seki et al., 2007*; *Saitou et al., 2012*; *Loda et al., 2022*; *Liu et al., 2025*). After they enter the gonad at E10.5, PGCs take on germ cell character (*Nicholls et al., 2019*), turn on DDX4 and initiate cyst formation while reprogramming is still ongoing. During E11.5, they induce Dazl, a major post-transcriptional mediator of early germ cell development, pluripotency re-establishment, and meiotic entry (*Ruggiu et al., 1997*; *Haston et al., 2009*; *Gill et al., 2011*; reviewed in *Fu et al., 2015*; *Zagore et al., 2018*; *Yang et al., 2020*). Even before meiotic entry, Dazl is needed to form stable cysts in zebrafish (*Bertho et al., 2021*) and in mice (*Rosario et al., 2019*).

The formation of germline cysts in mice is more complicated than in most animals. Mouse germline cysts arise from five rounds of mitotic divisions beginning at E10.5, so that each PGC generates almost 32 cells. However, the cysts break variably into an average of 4.8 smaller cysts before these divisions are completed at E14.5 when cells enter meiosis (*Lei and Spradling, 2013*; *Levy et al., 2024*). During meiosis, each smaller cyst slowly loses component cells, as every 2 days about one cell on average becomes activated to act as a nurse cell. The nurse cell transfers most of its cytoplasm into the residual cyst, shrinks to a small remnant, and undergoes programmed cell death (*Lei and Spradling, 2016*; *Niu and Spradling, 2022*; *Ikami et al., 2023*). In the pachytene substage of meiosis I, there is a substantial reorganization that brings mitochondria and ER sheets into close proximity by E18.5 (*Ruby et al., 1969*; *Nogawa et al., 1988*; *Pepling and Spradling, 2001*). Cyst breakdown is complete by 5 days after birth (P5) with the formation of 4–6 primordial follicles, each containing a single oocyte in which organelles largely transferred from nurse cells are gathered into an aggregate known as the Balbiani body (*Niu and Spradling, 2022*). Cyst polarity guides cyst fragmentation and oocyte specification, polarization, and organelle acquisition. Despite this, the only prior hint of a mouse fusome was a round structure that could be stained with certain antisera in PGCs and 2-cell cysts (*Hahnel and Eddy, 1987*; *Pepling and Spradling, 1998*).

Germline cysts also support germ cell rejuvenation, a universal process that removes accumulated damage each generation, allowing species to persist indefinitely. Germline rejuvenation takes place during meiosis in single-celled eukaryotes (*Unal et al., 2011*; *Suda et al., 2024*; *Xiao and Ünal, 2025*). Processes that rejuvenate organelles, such as mitochondria, restore rDNA copy number, and reset epigenetic chromatin marks have also been documented in meiosis during animal gametogenesis (*Cox and Spradling, 2003*; *Hill et al., 2014*; *Reik and Surani, 2015*; *Bohnert and Kenyon, 2017*; *Lieber et al., 2019*; *Pang et al., 2023*; *Yamashita, 2023*; *Xiao and Ünal, 2025*; *Spradling et al., 2025*). The Balbiani body in newly formed oocytes is associated with germ plasm formation and selection for mitochondrial function in many species (*Heasman et al., 1984*; *Kloc et al., 2004b*; *Marlow and Mullins, 2008*; *Spradling et al., 2022*; *Sekula et al., 2024*). Currently, however, germline rejuvenation events are thought to skew toward late oogenesis (*Bohnert and Kenyon, 2017*; *Xiao and Ünal, 2025*).

Here, we report that mouse PGCs and cyst cells elaborate a fusome-like structure that is rich in Golgi, endosomal vesicles, and ER associations. The mouse fusome distributes asymmetrically with stable microtubules that persist for part of the cell cycle. By the 8 cell stage, cyst cells reorganize into a rosette configuration that brings cells with multiple bridges (potential oocytes) and associated fusome material closer together. Developing cysts at this stage usually fragment non-randomly at sites lacking microtubule connections into six-cell cysts, implying that mouse ovarian cyst structure is more highly controlled than previously realized (*Lei and Spradling, 2013*; *Levy et al., 2024*). Beginning as PGCs and throughout cyst formation, endosome-Golgi associated degradation (EGAD)-mediated UPR pathway proteins including Xbp1 are expressed, suggesting that the oocyte proteome quality

control begins very early and continues. After pachytene, centrosome/Golgi-rich elements move to the oocyte and mediate the acquisition of organelles by the Bb. These results imply that a polarized fusome, whose properties have been significantly conserved in diverse animals including mammals, underlies female gamete development from the PGC to follicular oocyte stages.

## Results

### PGCs contain an asymmetric EMA granule

Early observations of a germ cell '(e)mbryonic (m)ouse (a)ntigen (EMA)' (*Hahnel and Eddy, 1987*) identified an 'EMA granule' in mouse primordial germ cells that resembles the *Drosophila* spectrosome, a fusome precursor (*Lin et al., 1994*; *Pepling and Spradling, 1998*). To investigate this relationship further, we determined that the EMA granule appears in PGCs as early as E9.5 and continues to be expressed in later cysts (*Figure 1A*, *Figure 1—figure supplement 1A*, *Figure 1—video 1*). We used lineage labeling to mark cysts (*Figure 1—video 2*) derived from single PGCs, and 3D surface rendering and volume quantification (*Figure 1—figure supplement 1B*), to demonstrate a significantly uneven distribution of the EMA granule within PGC daughter cells (*Figure 1B*). This asymmetry and the studies described below prompted us to name the EMA-rich structure the 'mouse fusome'.

The EMA-rich material reorganized and clustered together in female cysts by E13.5 (*Figure 1C*; *Figure 1—video 3*, *Figure 1—video 4*; *Figure 1—figure supplement 1C*), whereas in male cysts it remained separate and similar-sized in each cell (*Figure 1C'* and graph; *Figure 1—figure supplement 1D*), reminiscent of male differences in the *Drosophila* fusome (*Diegmiller et al., 2023*). In females, cyst regions with enriched fusome staining contained an average of 3 Tex14-positive bridges, whereas unenriched zones had an average of just 1 (*Figure 1D–D'* graph; *Figure 1—figure supplement 1E*). Thus, highly branched cells with more bridges, which are known to give rise to most oocytes (*Lei and Spradling, 2013*; *Lei and Spradling, 2016*; *Ikami et al., 2023*), contain more fusome. Similar clustering, known as rosette formation (*Figure 1H*), and increased fusome accumulation in pro-oocytes characteristically occur as germline cysts grow in *Drosophila* and other species (see *Hegner, 1914*; *Büning, 1994*; *Huynh and St Johnston, 2004*). Rosette formation is thought to prepare for sharing between nurse cells and oocytes.

### The mouse fusome is enriched in Golgi elements and vesicles

The fusome co-stained with the Golgi protein Gm130 (*Figure 1F*, *Figure 1—figure supplement 2H*) and the recycling endosomal protein Rab11a1 (*Figure 1F'*). Electron microscopy (EM) of mouse germ cells showed a prominent Golgi and endocytic vesicles in most E11.5 and E12.5 cyst germ cells in a subregion of similar shape to the EMA granule in light micrographs (*Figure 1E'*) consistent with a previous EM observations (*Jeon and Kennedy, 1973*; *Anderson and Beams, 1960*; *Clark and Eddy, 1975*). We found evidence that EMA granules require ongoing Golgi activity to be maintained. In E11.5 mouse ovaries cultured in vitro, treatment with the Golgi-inhibitor Brefeldin A for 6 hr caused EMA granules to shrink, but many reformed within 18 hr of its removal (*Figure 1—figure supplement 2I–I'*, *Figure 1—video 5*, *Figure 1—video 6*, *Figure 1—video 7*). The fusome often partly overlaps with ER (*Figure 1—figure supplement 2K–K'*). EM sections of E14.5 cysts revealed clusters of Golgi within adjacent cyst cells near ring canals, as expected for cysts after rosette formation (*Figure 1E and E''*). EMA-1 antisera react with multiple fucosylated glycolipids (*Apostolopoulos et al., 2015*; *Figure 1—figure supplement 2J–J'*). EMA staining disappears from germ cells at E14.5 (*Figure 1I*); however, very similar (but non-germ cell-specific) staining is also seen with wheat germ agglutinin (WGA) (*Figure 1G*) and this staining continues at E13.5 and later stages (*Figure 1G'*; *Figure 1—figure supplement 1F and G*).

### The mouse fusome overlaps centrosomes and associates with microtubules

A cell's microtubule cytoskeleton emanates from its centrosomes and from non-centrosomal microtubule organizing centers (ncMTOCs); it frequently controls trafficking of Golgi elements and positions them near centrosomes. To investigate whether the fusome is associated with microtubules, we stained E11.5 gonads with anti-acetylated tubulin (AcTub) and EMA. The fusome overlapped the centrosomes in interphase germ cells (*Figure 2A*), indicating it usually occupies a peri-centriolar

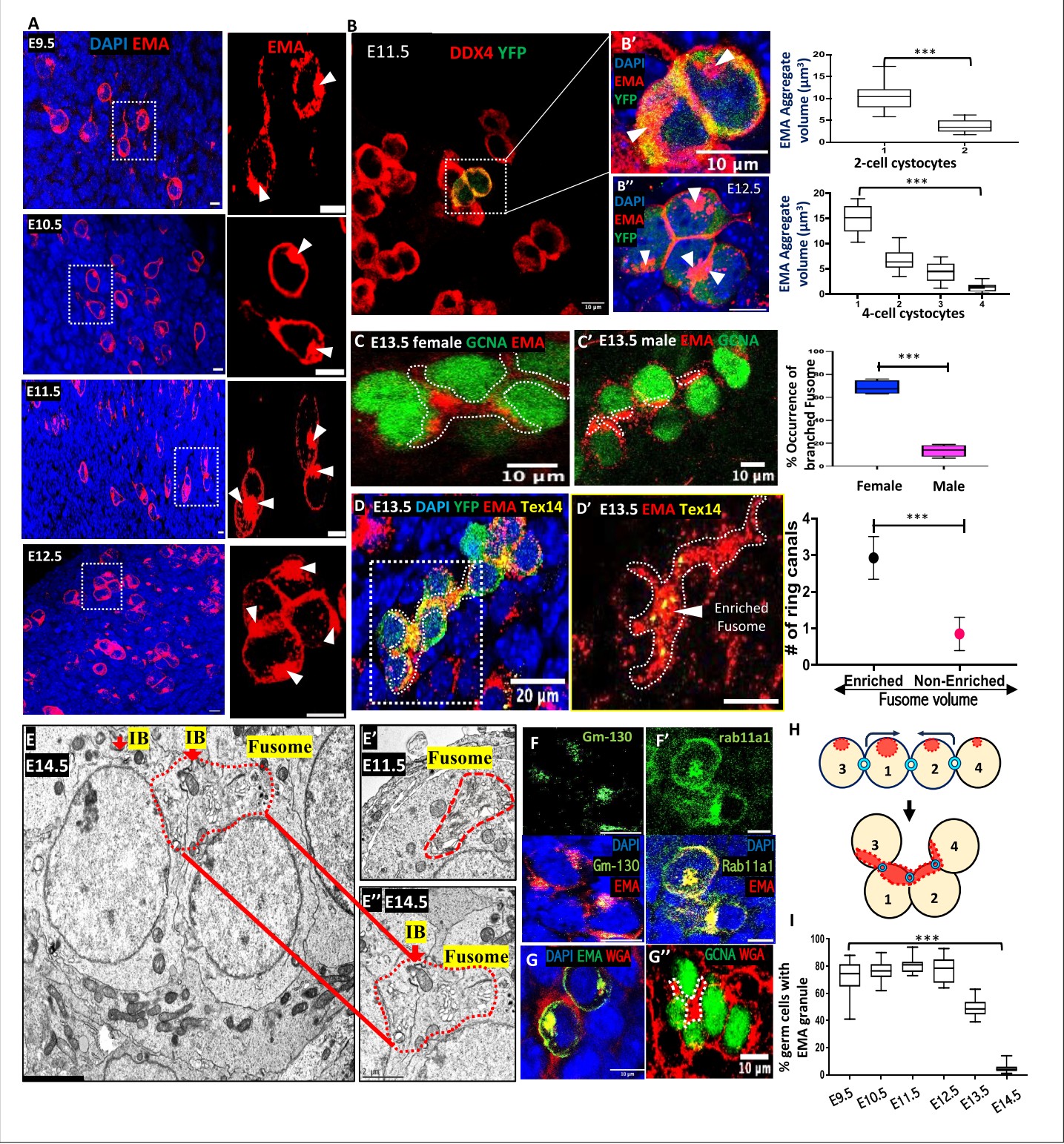

**Figure 1.** Mouse pre-meiotic primordial germ cells (PGCs) contain a 'fusome'. (**A**) PGCs and early germline cysts from E9.5-E12.5 ovaries. EMA (red), DAPI (blue). Boxed regions magnified at right (R) (arrows, EMA granules). (**B**) EMA granule asymmetry in an E11.5 2-cell cyst: Yellow cells represent a lineage-labeled 2-cell cyst marked with both YFP (green, lineage) and DDX4 (red). (**B′**) Boxed region showing EMA granules (white triangles). Graph at R: EMA granule volumes consistently differ between daughter cells in 2-cell cysts. N=16. (**B″**) Varying volumes of daughter cells within E12.5 4-cell lineage-labeled cysts. EMA granules (white triangles), EMA (red). Graph at R: EMA volume asymmetry in 4-cell cyst, N=18; (**C**) Rosette formation in E13.5 ovary and E13.5 testis (**C′**). GCNA (germ cell nuclei, green), EMA (fusome, red; outline, dotted white). Graph at R: % of female (blue) and male (red) cysts with

*Figure 1 continued on next page*

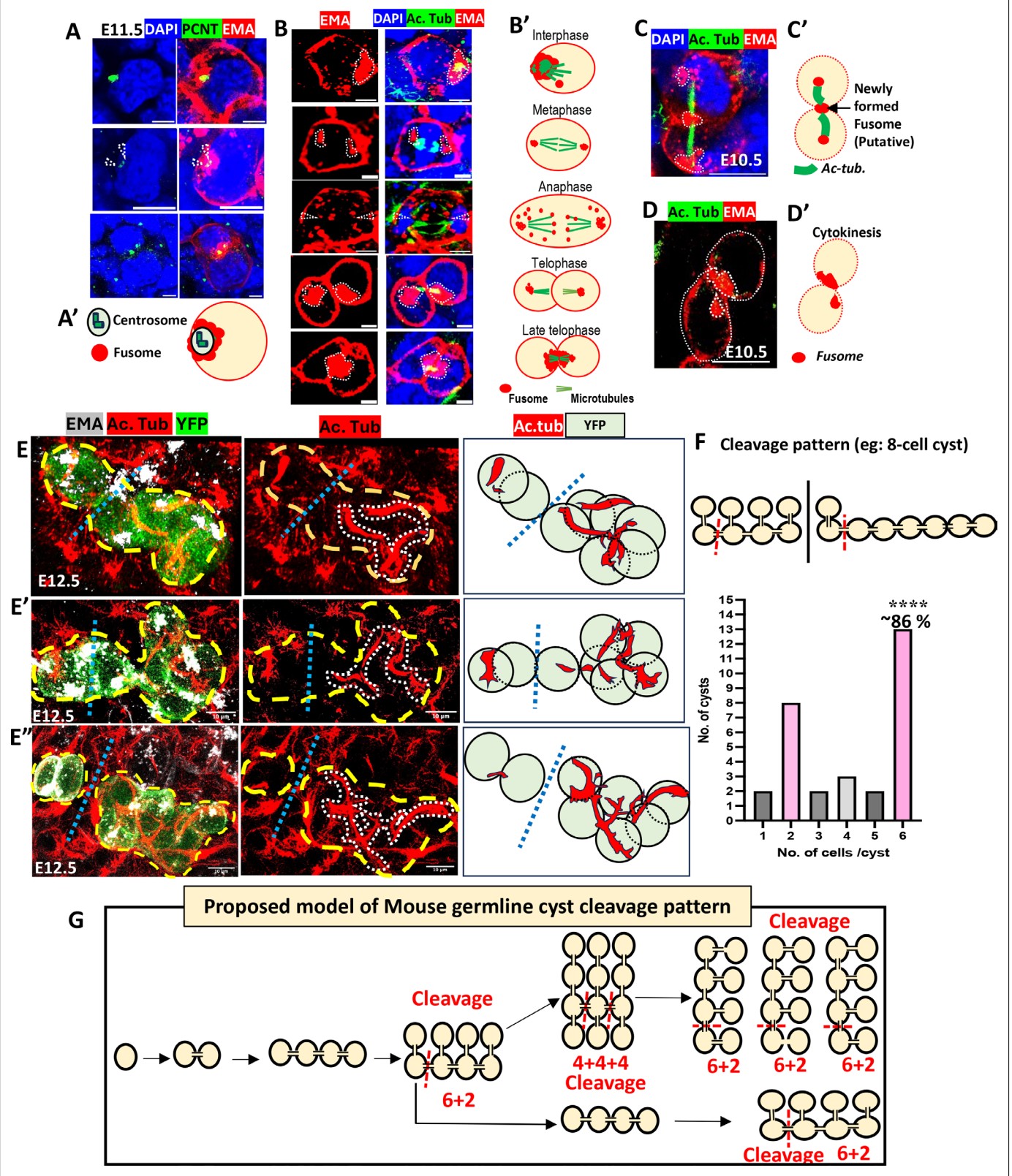

**Figure 2.** Stabilized spindle microtubules mediate fusome asymmetry and cyst breakage. (**A**) Pericentric fusome localization in E11.5 germ cells. The early fusome (EMA granule) associates with centrosomes (dashed arrowheads) in E11.5 germ cells: EMA (red), centrosomes PCNT (Pericentrin, green). Left column shows PCNT and DAPI alone. (**A'**) Summary. (**B**) Fusome (EMA) behavior during indicated stages of the cyst cell cycle. (**B'**) Diagrams summarize behavior at the listed mitotic stages deduced by AcTub (Acetylated Tubulin) staining. (**C, C'**) Symmetric early telophase fusome.

*Figure 2 continued on next page*

*Figure 2 continued*

(**D,D'**) Asymmetric fusome segregation during late cytokinesis. (**B–D**) EMA, acetylated tubulin (AcTub) and DAPI. (**E-E''**) Three lineage-marked (YFP) E12.5 8-cell cysts in early interphase stained to reveal microtubules (AcTub) and fusome (EMA). The absence of spindle remnants in one cyst region (gap in AcTub) predicts future cyst breakage (blue dashed line, summary at R only of YFP+ cells). The **E''** cyst has already broken into 2-cell and 6-cell cyst derivatives. (**F**) Frequency distribution predicted cyst breakage products by size based on 15 lineage-labeled cysts analyzed as in **E** (7-cell: 3; 8-cell: 8; 9-cell: 1; 10-cell: 3). Binomial test (see text) compared observed 6-cell cyst production frequency (13/15) to prediction for single random junction breakage of an 8-cell cyst. (****$p$<0.0001). (**G**) Model of cyst production and breakage into four 6-cell cysts and 4 2-cell cysts. Scale bars: 5 μm (**B**), 10 μm (**A, C–E**).

The online version of this article includes the following video and figure supplement(s) for figure 2:

**Figure supplement 1.** Validating microtubule-dependent fusome formation and its distribution during cyst fragmentation.

**Figure 2—video 1.** Fusome localization during interphase stage.
https://elifesciences.org/articles/109358/figures#fig2video1

**Figure 2—video 2.** Fusome localization during the mitosis stage.
https://elifesciences.org/articles/109358/figures#fig2video2

**Figure 2—video 3.** Fusome localization during the anaphase stage.
https://elifesciences.org/articles/109358/figures#fig2video3

**Figure 2—video 4.** Fusome localization during the telophase stage.
https://elifesciences.org/articles/109358/figures#fig2video4

**Figure 2—video 5.** Fusome localization during the late telophase stage.
https://elifesciences.org/articles/109358/figures#fig2video5

**Figure 2—video 6.** 3D visualization of fusome and acetylated tubulin connections within lineage labeled 8-cell cyst.
https://elifesciences.org/articles/109358/figures#fig2video6

**Figure 2—video 7.** 3D visualization of fusome and acetylated tubulin connections within lineage labeled 10-cell cyst.
https://elifesciences.org/articles/109358/figures#fig2video7

domain. In germ cells undergoing mitosis (*Figure 2B*), the fusome is associated with centrioles during interphase, but (typical for Golgi) dispersed as the mitotic spindle forms and for most of metaphase. In telophase, fusome vesicles gather at the spindle poles in both daughter cells and at the arrested cytokinesis furrow, which still retains the elongated spindle/midbody microtubules typical of cyst-forming cell cycles (*Figure 2B*). Small EMA-positive vesicles are often associated with these telophase spindles, providing evidence that fusome vesicles can move along microtubules (*Figure 2C*, *Figure 2—figure supplement 1D*). 3D summary movies of germ cells at each mitotic stage stained for AcTub and EMA are shown in *Figure 2—video 1*, *Figure 2—video 2*, *Figure 2—video 3*, *Figure 2—video 4*, *Figure 2—video 5*.

The mouse fusome was rarely seen at just one pole of a mitotic spindle, like the *Drosophila* fusome at early stages of GSC and cystoblast division (*de Cuevas and Spradling, 1998*). Some EMA-reactive material appears at the cytokinesis furrow before it is seen at the fusome (*Figure 2C*; *Figure 2—figure supplement 1B-C*). Asymmetry in the mouse fusome distribution arose only in late telophase. Clusters of fusome and centrosomes approached the arrested furrow (*Figure 2D*, *Figure 2—figure supplement 1B-C*). At this point, short microtubule arrays were visible (*Figure 2—figure supplement 1B*). Small fusome-containing vesicles are often observed in association with microtubules as shown in *Figure 2—figure supplement 1D*. Soon after, the amount of fusome in the two daughter cells becomes asymmetric, suggesting that differential activity or persistence of fusome transport late in cytokinesis generates asymmetry. Ovaries cultured briefly in vitro support continued germ cell division and fusome asymmetry. Fusome formation is severely affected in such cultures if microtubules (MTs) are disrupted by cold or by Ciliobrevin D treatment (*Figure 2—figure supplement 1E*).

## Cyst fragmentation is non-random and correlates with microtubule gaps

Mouse cysts produce an average of 4–6 oocytes per PGC as a result of initial cyst breakage into multiple, independently developing subcysts that each produce one oocyte (*Lei and Spradling, 2013*; *Niu and Spradling, 2022*). Production of uniform-sized oocytes, as observed, would seem to require programmed breakage into uniform subcysts (*Spradling et al., 2022*), but some random breakage also occurs (*Lei and Spradling, 2013*; *Ikami et al., 2023*; *Levy et al., 2024*). To assess the role of

*Figure 1 continued*

branched fusomes indicative of rosette formation (N=26 for each). (**D**) Ring canal abundance in fusome-enriched cells. A E13.5 lineage-labeled (YFP, green) cyst, fusome (EMA, red; outline, dotted white), and ring canals (TEX14, yellow). (**D'**) Zoomed image (boxed region in **D**) showing branched region with enriched fusome (white triangle) containing multiple ring canals. Graph at R: Ring canal number vs. fusome enrichment (≥10 μm³). N=54. (**E-E''**) EM of an E14.5 cyst in rosette configuration showing a Golgi-rich fusome spanning an intercellular bridge (**E''**). (**E'**) EM of an E11.5 PGC with a Golgi-enriched region (red outline) and likely EMA granule (compare to 1A-B). (**F-F'**) E11.5 germ cells with EMA granules (EMA, red) co-stained with the Golgi markers **F**. (GM130, green) or **F'** (Rab11a1, green). (**G**) Co-staining of Wheat germ agglutinin - WGA (red) and EMA (green) in E11.5 germ cells. (**G'**) WGA (red) staining of rosette fusome in E13.5 ovary: GCNA (nuclei, green). (**H**) Schematic of rosette formation in 4-cell cyst. (**I**) Plot showing EMA staining loss in germ cells after E13.5. (N=15 per stage). Student's t-test was used for each graph in Figure 1. (***p<0.001). Scale bars: 5 μm (**A, F, F'**), 10 μm (**B-B''**, **C-C'**, **D'**, **G-G'**), 20 μm (**D**), 2 μm (**E**).

The online version of this article includes the following video and figure supplement(s) for figure 1:

**Figure supplement 1.** EMA/Lectin-stained aggregate (Mouse fusome) distribution in pre-meiotic primordial germ cells (PGCs).

**Figure supplement 2.** EMA/Lectin-stained aggregate (Mouse fusome) distribution in pre-meiotic primordial germ cells (PGCs).

**Figure 1—video 1.** EMA staining disappearance from germ cell membrane in E13.5 ovary and acquiring a continuous branched appearance within the germline cyst.
https://elifesciences.org/articles/109358/figures#fig1video1

**Figure 1—video 2.** Successful single-cell lineage labeling of germ cells.
https://elifesciences.org/articles/109358/figures#fig1video2

**Figure 1—video 3.** Visualization of branched central EMA-stained fusome.
https://elifesciences.org/articles/109358/figures#fig1video3

**Figure 1—video 4.** Visualization of germ cell cluster specifically with branched fusome.
https://elifesciences.org/articles/109358/figures#fig1video4

**Figure 1—video 5.** Successful Golgi formation within germ cells of in vitro cultured fetal gonads.
https://elifesciences.org/articles/109358/figures#fig1video5

**Figure 1—video 6.** Validation of Brefeldin A (BFA) effect on Golgi formation.
https://elifesciences.org/articles/109358/figures#fig1video6

**Figure 1—video 7.** Reversibility of Brefeldin A (BFA) treatment.
https://elifesciences.org/articles/109358/figures#fig1video7

microtubules and the fusome in programming cyst fragmentation, we tracked AcTub and the fusome in lineage-labeled E12.5 cysts (*Figure 2E*, *Figure 2—figure supplement 1F*). In about 10% of interphase cysts, presumably those in very late telophase or early in the subsequent cell cycle, persistent microtubule arrays were observed that connected the cells in pairs (*Figure 2E*, *Figure 2—figure supplement 1F-H*). When we examined a total of about 50 large cysts, 15 larger cysts of 7–10 cells were found where the MTs in all cells could be analyzed completely. Three examples are shown in which projections of 8-cell lineage-labeled cysts are outlined (dashed lines). In the left column, lineage, the fusome and MTs are shown (*Figure 2E–E''*; *Figure 2—figure supplement 1H*). Unexpectedly, the microtubule connections (middle column) showed a large discontinuity of 10 μm or more between just two of the cystocytes (*Figure 2E*). Likewise, microtubule bundles in E' and E'' are also absent between two cells (straight dashed line *Figure 2E'-E''*, *Figure 2—figure supplement 1H*). The absence of an MT connection predicts cyst breakage, as observed in one of the cysts (*Figure 2E''*), where separation into a 2-cell and 6-cell cyst has already occurred. We also examined how the fusome distributes in these cysts (*Figure 2—figure supplement 1H*).

The patterns of cyst breakage predicted by MT absence were highly non-random (*Figure 2F*). A 6-cell cyst is predicted based on the position of the microtubule gaps in 13/15 8-cell to 10-cell cysts analyzed: see *Figure 2E*, *Figure 2—videos 6, 7* (movies of an 8-cell cyst, or 10-cell cyst, with cleavage diagram in *Figure 2—figure supplement 1G*); and *Figure 2—figure supplement 1H* (fusome

separation in the three cysts from *Figure 2E–E''*). A binomial probability calculation showed that the chance of obtaining a 6-cell cyst product in 13 of 15 trials following random breakage of a single junction in an 8-cell cyst (where 2 of 7 broken linkages produce a 6-cell product) was at most 4.53E-06.

## The mouse fusome associates with the polarity protein Pard3

Microtubules play a critical role in ovarian cyst formation and polarity in multiple species. Some of the most important regulators of MTs and polarity are the Par proteins that function in epithelial cell polarity, embryonic development, asymmetric neuroblast division (*Petronczki and Knoblich, 2001*; *Hapak et al., 2018*) and oocyte formation in *C. elegans*, *Drosophila* and vertebrates (*Kemphues et al., 1988*; *Huynh and St Johnston, 2004*; *Moore and Zernicka-Goetz, 2005*). *Drosophila* meiotic cysts express Par3/Baz and Par6, another apical Par complex protein, along with beta-catenin/Arm in a ring-like arrangement close to the ring canals (*Cox et al., 2001*; *Huynh et al., 2001a*; *Huynh et al., 2001b*; *Huynh and St Johnston, 2004*). This relatively small membrane zone likely represents the apical domain of cyst germ cells.

We investigated the expression of Pard3, the mouse ortholog of Par3/Baz, to determine if a similar arrangement of the highly conserved Par proteins occurs in mouse germline cysts. Pard3 expression localized around the fusome but extended beyond it, a pattern reminiscent of the larger ring of Baz in *Drosophila* cyst cells (*Figure 3A*, *Figure 3—video 1*). At E13.5, cysts in rosette configuration show Pard3 and the fusome enriched around a central cell (*Figure 3B*, dashed region from *Figure 3—video 2*) but not in E13.5 male cysts (*Figure 3—video 3*). This co-localization was validated in multiple lineage-labeled E13.5 cysts (*Figure 3D*; *Figure 3—figure supplement 1A–B*). Ring canals were concentrated in this zone (*Figure 3C*, *Figure 3—figure supplement 1C*). Hence, mouse cysts have ring canals and an apical Par protein in the same small anterior location as in *Drosophila*, and the mouse fusome occupies the corresponding zone as the *Drosophila* fusome. These findings suggest that mammals form and polarize oocytes using a conserved system found in diverse animals based on Par genes, cyst formation, and a fusome.

## The mouse fusome associates with Golgi proteins involved in UPR beginning at the onset of cyst formation

We performed single-cell RNA-sequencing (scRNA-seq) of mouse gonads to gather additional fetal germ cell data (see Methods) and combined them with data from *Niu and Spradling, 2022* to make an integrated dataset of more than 3000 germ cells spanning E10.5 to P5 (*Figure 3F–H*; *Figure 3—figure supplement 2D–H*). Our gene expression studies showed that PGCs and early cysts actively express many genes that enhance proteome quality by monitoring, refolding, or expelling misfolded or aggregated proteins for degradation. The UPR pathway associated largely with the ER in yeast is regulated by *Hac1*, the ortholog of *Xbp1*. We observed high levels of *Xbp1* (*Figure 3I*) and many of its target genes (*Figure 3I and I'*) that increase quality secretory protein production. Membrane proteins and lipids are also promoted by this pathway.

In animals, the UPR pathway has been expanded with several additional branches (*Mitra and Ryoo, 2019*; *Sicari et al., 2019Machamer, 2008*). We also observed expression of *Creb3l2* (ER/Golgi resident transcription factor), *Atf4* (General UPR transcription factor activated by Golgi stress), *Golph3* (Golgi phosphoprotein for membrane trafficking), and *Arfgap(s)* (involved in both Golgi to ER and Golgi to endosome trafficking) during mouse cyst formation (*Figure 3—figure supplement 2F*). Xbp1 activates genes, such as the chaperones (*Hsp5a/Bip, Hsp90b1; DNAjc10; Lman2; Dnajb9*), protein-disulfide isomerases (*Erp44, Pdia3, Pdia4, Pdia6, Creld2*), the Endoplasmic Reticulum Associated Degradation (ERAD) regulator *Syvn1/Hrd1* and the ERAD components (*Sec61b; Ubxn4, Psmc6, Ube2j2, Get4, Ufd1*) (*Taniguchi and Yoshida, 2017*). Thus, repair and expansion of ER, Golgi, and secretory components begins immediately or even before migrating PGCs reach the gonad (*Figure 3I*).

Xbp1 also supports synthesis of ER membrane phospholipids via upregulation of targets, such as *Elovl1*, *Elovl5*, and *Elovl6*, and *Chka*. Other highly expressed targets maintain ER and Golgi structure and are orthologs of *Drosophila* fusome proteins (*Figure 3I'*, *Lighthouse et al., 2008*). Thus, the adaptive UPR pathway and its branches are active during cyst formation and later in prophase as well (*Figure 3I–I'*).

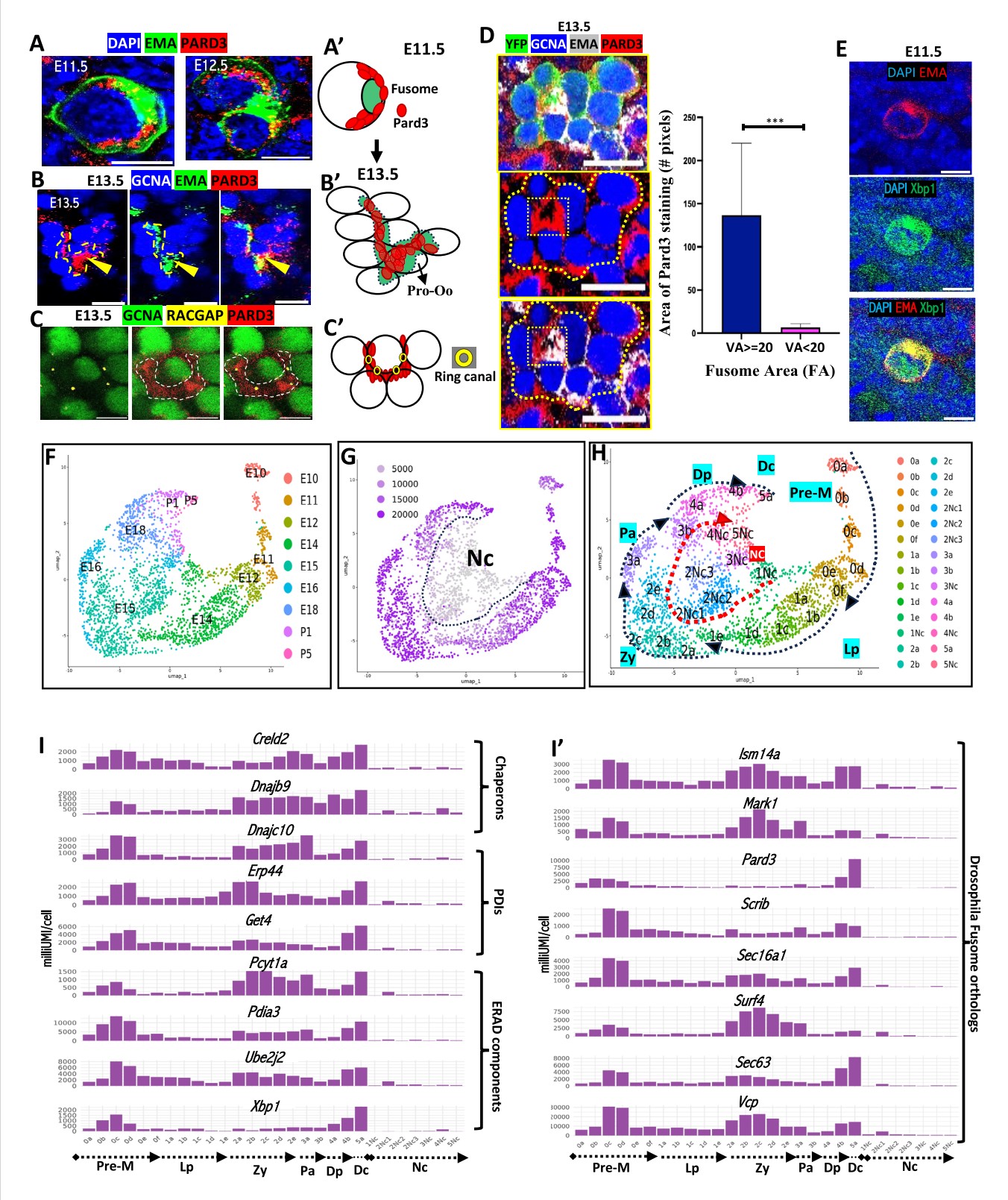

**Figure 3.** Mouse fusome associates with Pard3 and apical polarity. (**A–B**) Pard3 associates with fusome as observed in E11.5-E13.5; Gonad stained for Pard3 (red), EMA (green) and DAPI (blue) (**A, A'**) and after rosette formation at E13.5 (**B, B'**). (**C-C'**) Ring canals (RACGAP, yellow) localize within the Pard3+ (red) apical domain in germ cells (GCNA, green). (**D**) A lineage-labeled E13.5 cyst (YFP, green); channels below show enrichment of Pard3 (red) with enriched fusome (EMA, gray). Graph: Quantification of Pard3 stained area colocalizing with large- ≥ 20 µm² and small <20 µm² fusome

*Figure 3 continued on next page*

Figure 3 continued

within lineage labeled cyst (Student's t-test, N=13; ***p<0.001). (E) Xbp1 (green) enrichment in EMA (red) granule of E11.5 PGC. (F–H) scRNA-seq of E10.5-P5 gonad. UMAP of re-clustered germ cells at various stages (F), UMAP (G) UMI Feature Plot; NC = nurse cells. (H): UMAP with clusters labeled in ascending order of meiotic development. pre-meiotic (Pre-M), leptotene (Lp), zygotene (Zy), pachytene (Pa), diplotene (Dp), dictyate (Dc). (I-I') Bar plots: (I) Xbp1, Xbp1-target expression plots. (I') Genes orthologous to fusome components. Scale bars: 10 μm (A–C, E), 20 μm (D).

The online version of this article includes the following video and figure supplement(s) for figure 3:

Figure supplement 1. Pard3 gene expression in E12.5-E13.5 gonad.

Figure supplement 2. Validation of ScRNA-seq analysis to show mouse fusome association with Golgi-UPR pathway.

Figure 3—video 1. Pard3 staining pattern within pre-meiotic germline cyst.

https://elifesciences.org/articles/109358/figures#fig3video1

Figure 3—video 2. Pard3 and fusome as a continuous branched structure within female germline cyst.

https://elifesciences.org/articles/109358/figures#fig3video2

Figure 3—video 3. Pard3 and fusome as discontinuous, separate structures within male germline cysts.

https://elifesciences.org/articles/109358/figures#fig3video3

## UPR genes are active during cyst formation and depend on Dazl

We investigated how cyst development and fusome formation depend on Dazl. First, we confirmed that *Dazl^-/-* mice retard the PGC to germ cell transition (*Figure 4—figure supplement 1C–E*). *Dazl^-/-* E12.5 cysts showed higher levels of the DNA methylase Dnmt3a expression compared to controls (*Figure 4A*; *Figure 4—figure supplement 1A–B*), suggesting that germ cell DNA demethylation has slowed. Ring canals highlighted by staining for Tex14 were only half the size of wild-type canals at E13.5 or structurally abnormal, showing that cyst formation was slowed and cytokinesis was sometimes abnormal (*Figure 4B*).

We also performed scRNA-sequencing of *Dazl* female homozygous mutant gonads at E11.5 and E12.5 and re-clustered them with wild-type germ cells at these stages for comparison (*Figure 4C*). At E11.5, the merged germ cells cluster together, suggesting that *Dazl^-/-* has minimal effects on germ cells at E11.5, the stage it is first expressed (*Figure 4C*). However, E12.5 WT and *Dazl^-/-* germ cells cluster separately (*Figure 4C*). Wild-type and *Dazl^-/-* expressed similar amounts of EGAD-mediated UPR pathway genes at E11.5 germ cells (*Figure 4—figure supplement 1F*), whereas wild-type germ cells expressed lower levels compared to *Dazl^-/-* cells at E12.5 (*Figure 4C'*). Thus, UPR genes in *Dazl^-/-* cells act similarly to pluripotency genes and retain higher expression levels at E12.5 than wild-type.

The high expression of Xbp1 and many of its target genes in PGCs and early cyst cells argues that early germ cells are expanding ER and Golgi production and increasing membrane protein quality. To validate that XBP1 is active in germ cells, we employed an IRE1-XBP1 ratiometric assay which utilizes a genetically encoded dual-fluorescent reporter system (see Methods, *Figure 4—figure supplement 1I*). First, we performed magnetic-activated cell sorting (MACS) to purify SSEA1-labeled germ cells from the E11.5 gonad (*Figure 4—figure supplement 1G–H*). We delivered the IRE1-XBP1 sensor to SSEA1 +ve germ cells and SSEA1 -ve control somatic cells by transfection. Both types of cells fluoresce in proportion to IRE1-splicing activity which measures Xbp1 activation activity. The construct also constitutively expresses a fluorescent marker such that the color ratio will represent a quantitative measure of relative IRE1-XBP1 activity between cells and under different conditions. Our observations showed that IRE1-Xbp1 activity in E11.5 germ cells was significantly higher than in somatic cells (*Figure 4D*; *Figure 4—figure supplement 1I*). *Dazl* mutant cells showed persistent higher levels of Xbp1 activity at E12.5 (*Figure 4D'–D'''*).

The adaptive UPR pathway destroys misfolded proteins using proteasome activity, particularly the 20S proteasome (*Vembar and Brodsky, 2008*; *Figure 4H*). We measured proteasome activity within E11.5 PGCs to investigate whether the UPR pathway was paired with increased proteasome activity, indicating it served as an adaptive cellular mechanism of cell rejuvenation in PGCs destined to produce oocytes. Using a fluorogenic substrate to measure 20S proteasome activity, we compared MACS-sorted germ cells to somatic cells and found that germ cells have significantly higher proteasome activity than somatic cells at E11.5 (*Figure 4E*) and E12.5 (4E' Left). Proteasome activity continued to stay even higher in E12.5 *Dazl-/-* germ cells, suggesting stressful persistent activation of UPR keeping *Dazl-/-* germ cells in an arrested state (*Figure 4E'* Right, 4E").

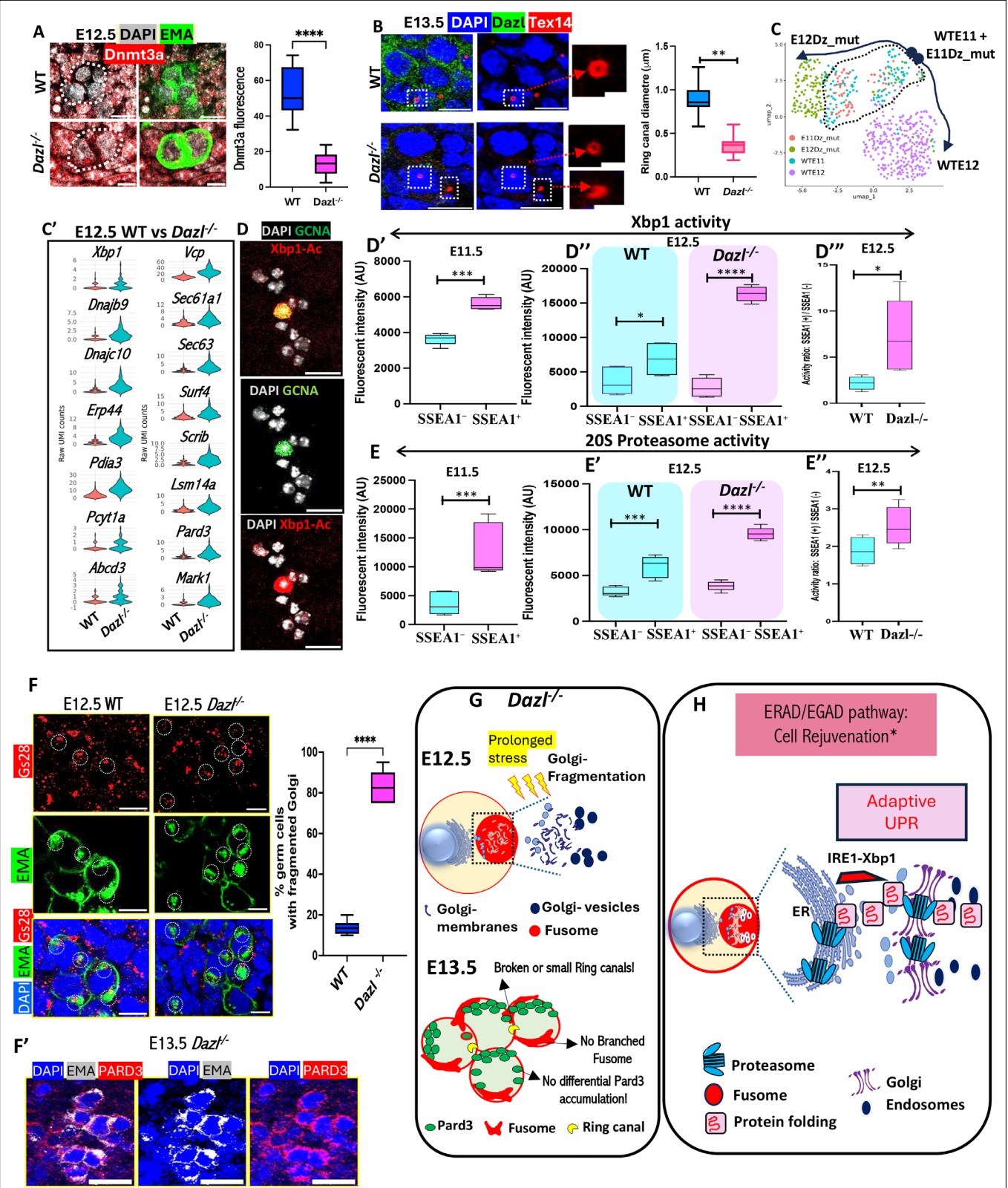

**Figure 4.** Unfolded protein response (UPR) genes are active during cyst formation and controlled by Dazl. (**A**) Dnmt3a and EMA levels at E12.5. Dnmt3a levels are reduced in wild-type (WT) compared to *Dazl⁻ᐟ⁻* germ cells. Graph - Dnmt3a fluorescent levels within germ cells as normalized with somatic cells in WT versus *Dazl* mutant gonad. (N=10 tissues; **p<0.05). (**B**) Ring canals are smaller and defective in E13.5 *Dazl⁻ᐟ⁻* cysts compared to WT. (N=44; **p<0.05). (**C**) scRNA-seq of E11.5 and E12.5 WT and *Dazl⁻ᐟ⁻* gonad germ cells. UMAP. Germ cell clusters overlapped at E11.5 and segregated at E12

*Figure 4 continued on next page*

*Figure 4 continued*

of WT and *Dazl^-/-*. (**C'**) Xbp1, Xbp1 targets, and fusome orthologs in WT vs *Dazl^-/-* germ cells. (**D**) Validation of IRE1-Xbp1 assay: Ovarian cells visualized by fluorescent microscopy showing GCNA labeled bigger germ cells with higher Xbp1 fluorescence than smaller somatic cells (**D'-D'''**) IRE1-Xbp1 assay comparing SSEA1+germ vs SSEA1– somatic cells at E11.5 and WT vs *Dazl^-/-* germ cells at E12.5. (**D'**; 6 experiments: ~32 mice, ≥5 mice, and ≥20 ovaries per experiment, **D''-D'''**; 3 experiments: ~40 mice, ≥5 mice, and ≥25 ovaries per experiments, *p<0.05, **p<0.01, ***p<0.005, ****p<0.0001) (**E-E''**) Proteasome activity comparing SSEA1+germ vs SSEA1– somatic cells at E11.5 and WT vs *Dazl^-/-* germ cells at E12.5. (N=3 biological assays with ~35–60 E11.5 ovary per assay and ~25–28 E12.5 ovaries were used per assay. *p<0.05, **p<0.01, ***p<0.005, ****p<0.0001) (**F**) Golgi fragmentation in E12.5 *Dazl^-/-* germ cells stained with golgi marker Gs28 (red), EMA (green) and DAPI (blue). Graph: germ cell percent with fragmented Golgi in wild-type versus *Dazl* mutant mouse gonad (Student's t-test: N=16, ***p<0.005) (**F'**) Failure of E13.5 *Dazl^-/-* germ cells to form EMA (gray) rosettes or enrich Pard3 (red). (**G**) *Dazl^-/-* effects on fusome, Golgi and Pard3. (**H**) Proposed function of fusome-mediated regulation of ERAD-UPR proteostasis. Scale bar: 10 μm (except zoomed in 2 μm).

The online version of this article includes the following figure supplement(s) for figure 4:

**Figure supplement 1.** Validation of female Dazl mutant phenotype, magnetic-activated cell sorting (MACS) and Activity assays.

We also studied the effects of *Dazl* mutation on the asymmetric enrichment of fusome in cells with multiple ring canals. Using both EMA and a Golgi marker, Gs28, we observed that germ cell Golgi begins to fragment in *Dazl^-/-* mutant germ cells even by E12.5 (*Figure 4F*, dotted circles, graph). High levels of Golgi stress documented by the increased IRE1-Xbp1 activity and elevated proteolysis in these cells may lead to such fragmentation. These problems were accompanied by a dramatic arrest in cyst polarity development. *Dazl^-/-* germ cells at E13.5 showed virtually no differential expression of Pard3 or enrichment of fusome within particular cyst cells (*Figure 4F'*). Finally, in *Dazl^-/-* P0 gonads, large cells destined to become oocytes do not appear (*Figure 5—videos 7; 8*). The effects of *Dazl-/-* on germline cyst development and meiosis are summarized in *Figure 4G*. These observations collectively indicate an essential function of the mouse fusome in temporal regulation of the EGAD-UPR pathway within germ cells (*Figure 4H*). The observed loss of polarity caused by *Dazl-/-* mutation can explain the failure of oocyte production and the sterility of *Dazl* homozygotes.

## The mouse fusome persists during organelle rejuvenation in meiosis

In *Drosophila*, starting at pachytene, the microtubule cytoskeleton changes its structure and polarity (*Cox and Spradling, 2006*). The new polarity, with microtubules running along the fusome such that minus ends cluster in the oocyte, prepares selected nurse cell organelles for transport into the oocyte and the Bb of a new primordial follicle. Mouse nurse cell centrosomes, mitochondria, Golgi, ER, and possibly other organelles also move from nurse cells to the oocyte between P0 and P4 in a microtubule-dependent process (*Lei and Spradling, 2016*; *Niu and Spradling, 2022*). Prior to or during these events, *Drosophila* mitochondria are selected and rejuvenated by programmed mitophagy to enhance functionality (*Cox and Spradling, 2003*; *Lieber et al., 2019*; *Palozzi et al., 2022*; *Monteiro et al., 2023*). It is not currently known if selection and rejuvenation take place before movement to the oocyte, or if a significant amount also takes place en route.

Because the fusome is directly involved in organelle movement to the *Drosophila* Balbiani body, we looked for changes in the fusome and in gene expression that might relate to organelle rejuvenation and movement to the oocyte. For the reasons explained previously (see text for *Figure 1G*), we used WGA as a fusome marker beyond stage E14.5. Fusome staining with WGA persists within meiotic germ cells at E17.5 (*Figure 5A*; *Figure 5—figure supplement 1A–B*). Fusome volume remained in proportion with germ cell ring canal number, as outer nurse cells (with one ring canal) transfer organelles and cytoplasm to cells with more ring canals located closer to oocytes (*Figure 5A*, graph; *Figure 5—video 1*). Pard3 content also showed such a proportion (*Figure 5A'* graph). At E18.5, cyst development in the medulla toward wave 1 follicles diverged from wave 2 follicles in the cortex that become quiescent primordial follicles (*Figure 5B*; *Yin and Spradling, 2025*). Wave 1 oocytes grow substantially in volume (*Figure 5—video 2*, *Figure 5—figure supplement 1C*), enriching fusome and Pard3 content due to cytoplasmic transfer from surrounding nurse cells which decrease in size (*Figure 5B–B'*, graph, *Figure 5—video 3*, *Figure 5—video 4*). Oocyte enrichment of the fusome and Pard3 was confirmed using lineage-labeled cysts at P0 (*Figure 5C and D*). Such enrichment was significantly reduced in Dazl +/-heterozygotes (*Figure 5E* graph, *Figure 5—figure supplement 1*; *Figure 5—video 5*; *Figure 5—video 6*).

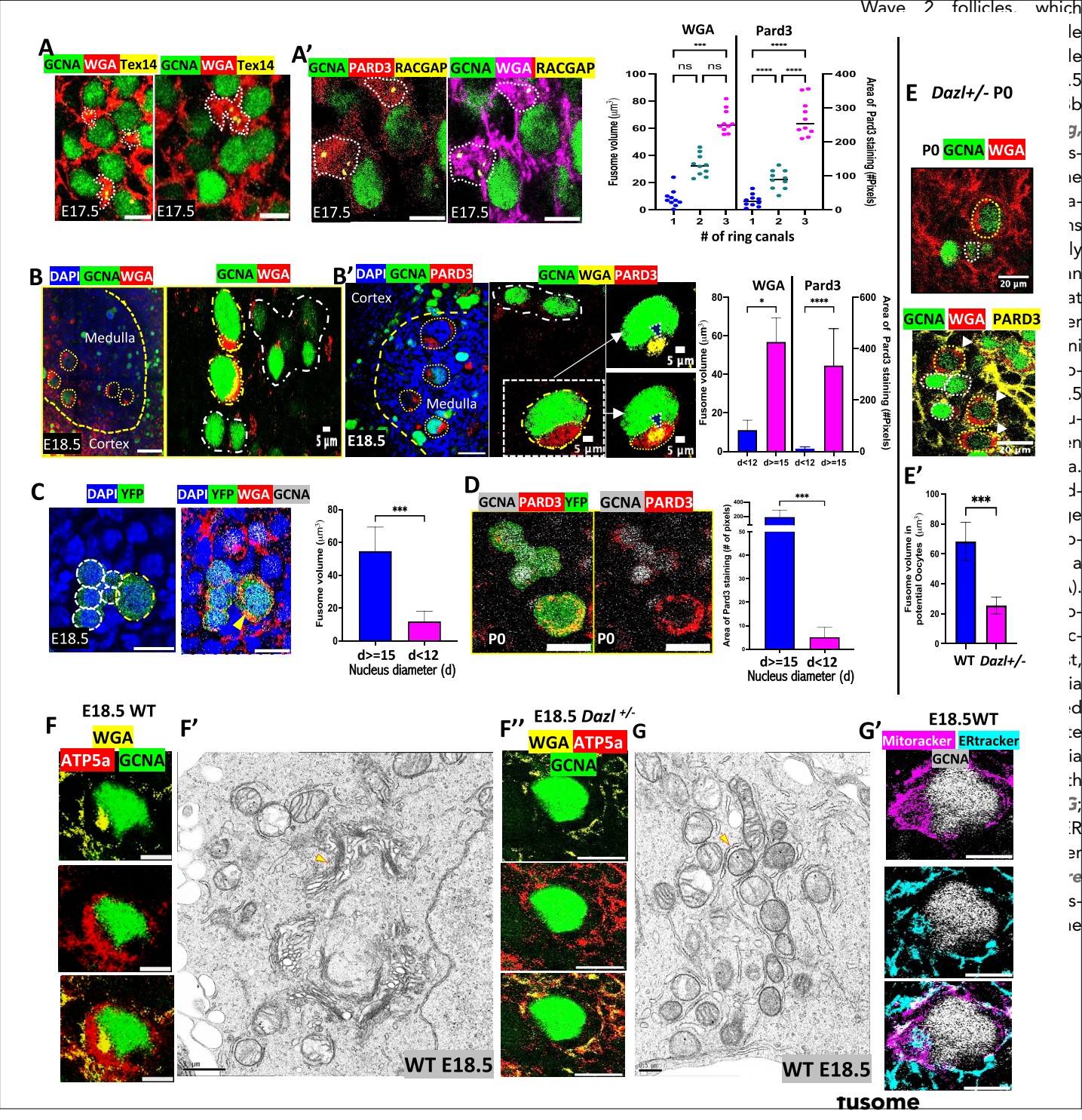

Wave 2 follicles, which

**Figure 5.** Fusome and Pard3 associate with endoplasmic reticulum (ER) and mitochondria prior to Balbiani body formation. (**A**) E17.5 ovary stained for WGA, GCNA, and TEX14. (**A'**) E17.5 ovary stained for GCNA, RACGAP, and PARD3; Graph: Fusome volume and Pard3 Stained area versus ring canal number N=65 (Fusome volume; N=51 (Pard3); ANOVA, ***p<0.005, ****p<0.0001). (**B-B'**) E18.5 ovary shows WGA-Fusome/PARD3 enrichment in large medullary oocytes vs smaller nurse cells; line: medulla/cortex boundary; dotted circle: large medullary oocytes; white dotted area: small nurse cells. The area marked as a white dotted rectangle is shown as a zoomed inset (white arrow). Black arrow in inset: WGA stained fusome; Graph compares fusome volume and Pard3 stained area versus Germ cell nucleus diameter (N=54 (WGA), N=37(PARD3); Student's paired t-test, *p<0.05, ****p<0.001). (**C**) Single cell lineage labeled E18.5 ovary stained for YFP, DAPI, WGA, and GCNA Graph: Within single-cell

*Figure 5 continued on next page*

One of the earliest and most highly conserved steps in animal gamete development is the formation of polarized germline cysts (*Giardina, 1901*; *Wilson, 1925*; *Büning, 1994*; *Matova and Cooley, 2001*; *Lu et al., 2017*; *Gerhold et al., 2022*; *Spradling et al.,*

*2022*). In the ovaries of most species, cysts become polarized during the synchronous mitotic cycles whose incomplete cytokinesis connects daughters via characteristic ring canals. In *Drosophila*, germline cyst divisions also produce and asymmetrically distribute a microtubule and vesicle-rich fusome whose polarity is essential to specify oocytes in addition to nurse cells. We found that mouse germline cysts also contain a similar organelle, the 'mouse fusome.' As in *Drosophila*, the mouse fusome starts as a small, membrane-rich cytoplasmic region within migrating PGCs before they arrive at the gonad. The nascent organelle propagates asymmetrically during synchronous, incomplete cyst divisions, involving stabilized spindle microtubules rich in AcTub and a program of arrested cytokinesis that produces distinctive ring canals. Stainable fusome MTs persist for only part of each cell cycle, but polarity is maintained since over five successive mitotic cycles the cyst undergoes rosette formation, predicts cleavage into smaller cysts, and concentrates fusome material in future oocytes. Finally, the mouse fusome associates in a distinctive manner with the apical Par complex located near the clustered ring canals, as originally described for the fusome in *Drosophila* cysts (*Cox et al., 2001*; *Huynh et al., 2001a*; *Huynh et al., 2001b*). Apical-basal patterning of *Drosophila* oocytes differs from many epithelial cells (review *Huynh and St Johnston, 2004*), but our studies show that both mouse and *Drosophila* oocytes localize the fusome and ring canals in a similar relationship to the apical Par domain.

## How conserved are fusomes in distant animal groups?

The recent detailed analyses of fusomes in Xenopus (*Davidian and Spradling, 2025*) and here in mice provide an opportunity to compare fusomes over a wider range of animals than was possible when well-characterized fusomes were limited to Holometabolous insects. Based on these examples, the most highly conserved aspects of polarized cyst production are the cell cycle changes that arrest telophase, stabilize the midbody, and lead to incomplete cytokinesis (*Gerhold et al., 2022*; *Price et al., 2023*). Somehow, this process polarizes the microtubules that arise in the fusome segment within the nascent ring canal and becomes a building block in growing cyst polarity. A longstanding mystery regarding fusome microtubule polarity is its exceptionally high sensitivity to microtubule inhibitors like colchicine, which appears to be much greater than cell division generally, or other developmental processes requiring microtubule-dependent vesicle transport (*Koch and Spitzer, 1983*). The sensitivity continues later in meiosis, since treating E17.5 mouse ovary cultures with just 10 nM colchicine strongly reduced Balbiani body formation but had no measurable effect on cell division (*Lei and Spradling, 2016*). Interestingly, mouse cysts develop asymmetry during a later period in cytokinesis than *Drosophila* cysts (*Figure 2C–D*). In Xenopus, another fusome-related property, cyst-wide divisional synchrony, showed more variation between sister cells than in tightly coordinated *Drosophila* cyst cells (*Davidian and Spradling, 2025*).

Qualitative differences in fusome structure and behavior have evolved between major animal groups. *Drosophila* fusomes and those of related insects are stabilized by an essential, proteinaceous skeleton containing alpha-spectrin, tropomodulin, and germline-specific isoforms of adducin-like proteins generated by differential splicing and cleavage of a polyprotein precursor produced by the Hts gene (*Petrella et al., 2007*). This skeleton may have evolved to control access to the fusome and hence to the oocyte by intracellular parasites, such as Wolbachia bacteria. An alpha-spectrin-containing skeleton was not detected in Xenopus fusomes (*Davidian and Spradling, 2025*) or in these experiments in mouse. The Xenopus fusome has robust microtubules that may independently control access. This structural difference may explain earlier failures to identify fusomes in vertebrates using orthologs to *Drosophila* fusome proteins like alpha-spectrin.

## Association of the mouse fusome with Golgi

The major role apparently played by Golgi in the mouse compared to the *Drosophila* or Xenopus fusomes is potentially significant. In contrast, the *Drosophila* fusome contains abundant ER cisternae, whose presence may facilitate lipid biosynthesis needed to produce high levels of new plasma and organellar membranes. Not only does rapid cyst formation demand high amounts of fresh membranes, but rejuvenation of new germ cells may require that their lipids and proteins be newly synthesized from simple building blocks (see *Spradling et al., 2025*). Mouse cysts grow more slowly and may associate extensively with Golgi because of an early high demand for protein secretion. In mouse, the zona pellucida directly contacts the oocyte and begins to be produced and secreted by the oocyte

in primary follicle oocytes that have just begun to develop. In contrast, the analogous *Drosophila* egg coating, the vitelline membrane, is produced by somatic follicle cells, and only starting in more mature follicles. Even primordial follicle oocytes contain abundant Golgi and multivesicular bodies near the Balbiani body (*Niu and Spradling, 2022*). Thus, the fusome's apparently stronger association with the Golgi in mouse but with the ER in *Drosophila* probably just reflects differences the relative timing and speed of events in oogenesis that take place in both organisms. This highlights the need to study fusome behavior throughout oogenesis and in the context of ongoing developmental and rejuvenation activity.

## Mouse cysts develop multiple uniform oocytes using a novel specific-cleavage mechanism

Mouse PGCs generate 4–6 oocytes using a multi-step mechanism involving synchronous incomplete cyst divisions, limited fragmentation into smaller cysts (involving <10% of cell-cell junctions) and continuous, slow nurse cell turnover (*Lei and Spradling, 2013*; *Lei and Spradling, 2016*; *Niu and Spradling, 2022*). Despite their complex pathway, mouse oocytes in newly formed primordial follicles appear quite uniform in size and are efficiently produced (*Spradling et al., 2022*), suggesting that a majority develop from final cysts of similar size and nurse cell content. However, lineage analyses showed many cyst cells move apart at least transiently. Live imaging studies revealed active movements likely responsible for cell displacement, as well as some cell detachment caused by random breakage (*Levy et al., 2024*).

Our studies of mouse fusome microtubules indicated that oocyte uniformity is promoted by programmed cyst breakage during mitotic divisions into cysts of uniform size (*Figure 2G*). AcTub arrays in cysts of 7–10 cells frequently contained a single large gap lacking MTs between a group of 6 cells and the other residual cells. Such gaps in a major cytoskeletal element of germ cells subjected to significant tissue movements would likely become breakage sites (*Figure 2E''*). Previous experiments also supported a special role for 6-cell cysts. We conclude that germline cyst breakage is largely programmed and efficient 6-cell cyst production explains the high frequency of synchronous six-cell mitoses observed during E11.5-E13.5 (*Pepling and Spradling, 1998* and directly mapped 6-cell cysts in young cyst stages prior to E14.5 *Lei and Spradling, 2013*). Efficient cell turnover and transfer of most nurse cell contents from 6-cell cysts into the oocyte probably explains the 5.1-fold increase in Golgi, 4.9-fold increase in mitochondria and fivefold increase in the number of accumulated centrosomes measured in P4 oocytes compared to E14.5 germ cells (*Lei and Spradling, 2016*).

How might growing cysts be programmed to acquire a gap in remnant spindle microtubule bundles affecting a specific junction? One possibility is the uneven distribution of the fusome in four-cell cysts (*Figure 1B''*). If the fusome is involved in stabilizing and associating with spindle remnants, shortly after M phase, the cell with the lowest fusome amount might fail acquire a spindle remnant of the necessary size at the next division, leading to 6:2 breakage. Involvement of the microtubule cytoskeleton in asymmetric divisions has been observed in many other situations (*Fichelson and Huynh, 2007*; *Kaltschmidt and Brand, 2002*; *Sunchu and Cabernard, 2020*; *Meiring et al., 2020*; *Planelles-Herrero et al., 2022*; *Watson et al., 2023*). Significantly, other sources of variation likely also occur at a lower level. Synchronously dividing 8 cell and 16 cell cysts are readily detected in low numbers (*Pepling and Spradling, 1998*). These large cysts may represent a specific alternative pathway that may preferentially take place in cysts within one of the three follicle waves that occur in mouse (Yin and Spradling, 2025). The model in *Figure 2G*, also predicts production of 4 2-cell cysts which were commonly observed prior to E14.5 and the onset of substantial nurse cell transfer (*Lei and Spradling, 2013*, Supplement).

## Dazl regulates cyst formation, fusome polarization, and oocyte specification

Female gamete development in many organisms is extensively controlled at the post-transcriptional level by RNA-binding proteins (*Mercer et al., 2021*; *Conti and Kunitomi, 2024*). In vertebrates, the RNA-binding protein Dazl is widely conserved and controls multiple aspects of early germ cell development by regulating a large number of target transcripts (*Zagore et al., 2018*). In mice, Dazl modulates the ongoing reprogramming of newly arrived gonadal PGCs to pluripotency beginning at E11.5 when it turns on *Gill et al., 2011*; *Nicholls et al., 2019*; *Haston et al., 2009*. Normally, Dazl

continues to function at later stages of oocyte development (*Conti and Kunitomi, 2024*), but *Dazl* mutants fail to enter or progress in meiosis.

Our studies defined a series of new functions for Dazl during cyst formation. We confirmed that pluripotency reactivation is slowed in *Dazl* mutants since DNA demethylation was delayed, and pluripotency genes failed to start downregulating at E12.5. Homozygous *Dazl* mutant females failed to normally regulate UPR genes, leaving their mRNA levels elevated. This likely resulted from a failure to downregulate their Xbp1 activator, since both Xbp1 activity and 20 S proteasome activity was also elevated in *Dazl-/-* germ cells. Overactivation of the UPR pathway appeared to cause Golgi fragmentation. These effects resembled the general slowing of normal developmental progression, as with pluripotency downregulation. We also observed abnormal cyst formation with some small and defective ring canals, as in zebrafish *Dazl* mutants (*Bertho et al., 2021*), but cysts remained intact and continued to develop. This allowed us to document that Dazl is needed for cysts and their fusomes to become polarized. In Dazl mutants, cysts did not undergo rosette formation or concentrate the fusome or Pard3 in cells with multiple ring canals. Oocytes apparently could not be specified and did not form. These results show that Dazl acts as a regulator that is essential for the many important aspects of cyst formation and oocyte production.

## Germ cell rejuvenation is highly active during cyst formation

Germ cells have long been known to propagate species without know limits over evolutionary time by rejuvenating gametes each generation in association with meiosis (*Weismann, 1892*; *Kirkwood, 1987*). Recently, the detailed cellular and molecular genetic mechanisms that make this possible both in single-celled eukaryotes and in animals have generated increasing interest (*Cox and Spradling, 2003*; *Unal et al., 2011*; *Chen et al., 2014*; *Bohnert and Kenyon, 2017*; *Lieber et al., 2019*; *Palozzi et al., 2022*; *Spradling et al., 2022*; *Yamashita, 2023*; *Xiao and Ünal, 2025*). By studying the fusome, which is already present in migrating E9.5 PGCs, and by sequencing mouse germ cells from the earliest stages of ovarian development beginning at E10.5, we gained new insights into the timing and processes that likely contribute to germ cell rejuvenation during mouse oogenesis.

These studies made it clear that multiple rejuvenation mechanisms are highly active from the earliest times of germ cell development (*Figure 3I*). In animals, the UPR pathway has been modified by the addition of several additional pathways besides the Ire1-Xba1 branch, all of which were expressed in early germ cells. Creb3l2 controls a transcription factor that increases production of COPII vesicles that carry out ER to Golgi transport and maintain a balance between secretory supply and demand. Insufficient matching can lead to Golgi stress (*Yin and Spradling, 2025*), which can be managed by activating the ATF4 pathway branch.

Expression of *Xbp1* mRNA or even Xbp1 protein using a general antibody is not sufficient to show Xbp1 activity, since this is controlled by Ire1-mediated splicing to produce the active Xbp1s isoform. We were able to purify enough E11.5 and E12.5 germ cells to carry out biochemical assays and show that mouse female germ cells have substantial levels of Xbp1s activity at both times, higher than in somatic cells. The high expression of many known Xbp1 targets also documents the high activity of this pathway (*Figure 3I and I'*). Early germ cells also express genes suggesting that they are expanding membranes via lipid biosynthesis and secretory pathway activity. Consistent with active adaptive-UPR rejuvenation of their proteomes, we documented substantial levels of active proteolysis in early germ cells at both E11.5 and E12.5.

The observations reported here suggest that EGAD—an emerging proteostatic mechanism—plays a previously unappreciated role. EGAD complements the well-established ERAD pathway by facilitating the clearance of misfolded proteins via Golgi-endosome trafficking and proteolysis, rather than removal from the ER and degradation in the proteasome as in ERAD. We observed substantial amounts of ERAD based on the expression of *Xbp1* and its many targets, and its documented activity and its endpoint 20 S proteasome activity. It is hard to quantitatively compare amount of EGAD activity, but the prominent role of the Golgi throughout the stages of mouse oogenesis and its location at the core of the Balbiani body suggests that EGAD fulfills a unique role in oocyte production. Additionally, oocyte EGAD must be capable of operating at the high level of proteostatic activity needed for germline rejuvenation without generating damaging and self-limiting side effects.

Rejuvenation genes were also highly expressed later in meiosis during zygotene and pachytene, in diplotene when mitochondria/ER contact is extremely high (*Figure 5G*), as well as in primordial

follicles (*Figure 3I and I'* 'Dc'). The early appearance of the fusome and its association with Golgi and ER-related processes that are central to adaptive UPR suggests that in animals, rejuvenation begins earlier in the generational cycle than it does in single-celled eukaryotes. Animal germ cells return to a pluripotent state by the end of cyst formation. Cyst formation precedes meiosis and represents an opportunity to enhance the rejuvenation process (*Spradling et al., 2025*). Our findings of early rejuvenation activity align with those of *Palozzi et al., 2022*, who showed that mitochondrial rejuvenation during oogenesis uses a special germline mitophagy and begins at the onset of meiosis. In yeast, mouse, and *C. elegans* oocytes, damaged materials are destroyed by lysosome-like degradation in mature oocytes and at meiosis II (*Bohnert and Kenyon, 2017*; *Zaffagnini et al., 2024*; *Xiao and Ünal, 2025*), similar to the lysosome-like turnover of remnant nurse cells and residual fusome contents near the onset of follicle formation in mouse and *Drosophila*. Thus, animal oocytes may engage in rejuvenation activities from before meiosis begins until meiosis II is completed.

In sum, our experiments argue that mouse germline cysts are produced by an ancient conserved process that modulates spindle microtubules to generate an asymmetric polarized fusome associated with apical Par proteins that enrich in oocytes. Proteins and processes involved in cellular rejuvenation are highly active throughout meiotic prophase and begin even in migrating PGCs. Germline cyst polarity ensures that organelles gathered from the nurse cells are delivered to the oocyte where they form a Balbiani body. In *Drosophila*, germline cysts, fusomes and Par-mediated polarity are all essential for oocyte production. The parallels shown here across 500 million years of evolution strengthen the case that germline cysts and polarized fusomes are likely to have been maintained in evolution because they contribute to the existential task of rejuvenating germ cells each generation.

## Materials and methods
### Experimental model and subject details

Mouse strains used - C57BL/6, CAG-cre/ERT2 mice, R26R-EYFP (as described in *Lei and Spradling, 2016*) and *Dazl*$^{+/-}$ (Strain #035880 from JACKSON laboratories). Genotyping was performed according to protocols provided by the JAX Genotyping was performed according to protocols from the JAX Mice database. Animals were provided with a proper light-dark cycle, temperature, humidity, food and water. Sexually mature females and males (6–8 weeks old) were kept for mating. Mating was confirmed through the observation of a vaginal plug. As per standard protocol, we designate the midday of the corresponding day as E0.5, which marks the beginning of embryonic development. Fetal gonads were dissected during E10.5-E18.5 as needed. The day pups are born is designated as P0 and were dissected for some experiments. The Institutional Animal Care and Use Committee of Carnegie Institution of Washington approved procedures for animal handling and experimentation.

### Single germ cell lineage labeling

The protocol used for single germ cell lineage labeling was performed as described previously (*Lei and Spradling, 2016*). Briefly, to obtain fetuses for lineage marking, adult female R26R-EYFP mice were mated with male CAG-cre/ERT2 mice. Tamoxifen was dissolved in corn oil (Sigma) and a single dose was injected intraperitoneally at 0.2 mg per 40 g body weight into pregnant female R26R-EYFP mice at E10.5. Fetal gonads/pups were dissected as needed in between E11.5-E18.5/P0. To analyze the lineage-labeled germ cells, fixed and frozen whole gonads were subjected to whole mount staining for chicken-GFP, the number of lineage-labeled germ cells of each ovary was determined by examining optically sectioned Z-stack images generated by confocal imaging of the entire ovary.

### Immunostaining

Dissected fetal gonads/ovaries were washed in phosphate buffer saline (PBS) and fixed immediately in cold 4% paraformaldehyde overnight at 4$^0$ C. Following the fixation, the gonads were washed three times with PBST$_2$, which is a mixture of phosphate-buffered saline (PBS) with 0.1% Tween 20 (Sigma) and 0.5% Triton X (Sigma). Each wash cycle lasted for 30 min. Next, gonads were incubated with primary antibodies mixed with blocking solution (PBST$_2$+10% Normal donkey serum) overnight at room temperature. The following day, gonads were incubated with fluorescein-conjugated secondary antibodies (Donkey Anti-Rabbit/mouse/rat Alexa-fluor 488/568/647, Invitrogen) overnight at room temperature. The next day, gonads were washed with PBST and stained with DAPI (Sigma) to visualize

nuclei. Gonads were mounted on slides with a mounting medium (Vector Labs) and analyzed using confocal microscopy (STELLARIS 8 DIVE Multiphoton Microscope, Leica). Germ cell numbers are quantified in each ovary by manually counting EMA/DDX4 stained germ cells in multiple 0.45 µm optical sections from different ovaries.

## 3D reconstruction

Three-dimensional model images and movies were generated by Imaris software (Bitplane). For z-stack images, different visualization options in the Surpass mode of Imaris helped to gain visualization control of the objects. To perform surface rendering and volume quantification using Imaris software, we imported your 3D image dataset for EMA/WGA staining. We started with a 'Volume' rendering for channels corresponding to EMA/WGA staining and then we adjusted the contrast, brightness and transparency in the 'Display Adjustment' window. Next, we used the Surfaces module to create surface renderings by selecting the appropriate fluorescence channel, applying manual thresholding and adjusting smoothing parameters to define the structure accurately. Once the surface is rendered, we navigated to the statistics tab to access volume measurements. Further visualization of the results was performed in 3D or orthogonal views to confirm accuracy and adjusted rendering properties, such as color or transparency as needed. To create spots for easy counting of labeled cells, we used the 'Spots' icon option in the 'Creation' menu and started the spot creation wizard. We set the approximate spot size as per the requirement. Next, we selected the 'Manual Creation' option to place spots individually. Each click placed a single spot at the corresponding position.

## Sexing and genotyping

Tail samples from mouse were collected and subjected to lysis and DNA extraction using Proteinase K-based lysis buffer treatment at 55 °C overnight followed by 95 °C incubation for 10 min. The samples were then centrifuged at 14,000 rpm for 5 min to pellet the debris. Extracted genomic DNA from supernatant was used directly and amplified with the Uba1, Sly and Zfy primer pairs for sex determination (*McFarlane et al., 2013*, Appendix 1-Key resources table). PCR reactions were performed using KAPA Fast Hotstart ReadyMix with dye (Kappa Biosystems, #KK5608) using the following PCR parameters: initial denaturation at 95 ° C for 5 min, 35 cycles with 94 °C for 30 s, 60 ° C and 72 ° C for 30 s, followed by final elongation at 72 ° C for 5 min. PCR products were analyzed with a DNA ladder (100 bp) on 2% agarose gels and visualized with ethidium bromide under UV-illumination. The male and female amplicon can be distinctly visualized due to different amplicon sizes of PCR products (see Primer details).

## Electron microscopy

Whole ovaries were dissected in PBS and fixed in 4% paraformaldehyde overnight. After three 3 min washes in cacodylate buffer, ovaries were postfixed in 1% OsO4, 0.5% K3Fe {CN}6, in cacodylate buffer for 1 hr and were rinsed twice for 5 min in cacodylate buffer and once for 5 min with 0.05 M maleate (pH 6.0). The ovaries were stained in 0.5% uranyl acetate overnight at 4 degrees rinsed in water, and dehydrated through an ethanol series. Following two 10-rain washes with propylene oxide, the ovaries were infiltrated with resin. The resin-embedded specimen was polymerized by incubation at 45 °C and $70^0$ C for 12 hr each. Silver-gold sections were cut, stained with lead citrate, and observed in the electron microscope.

## Single-cell RNA sequencing of wild-type E10.5, E11.5, and E15.5 gonad

Embryonic ovaries were dissected such that individual fetal gonad at E10.5 (18 gonads from 3 females), E11.5 (12 female gonads from 3 females) and E15.5 (10 gonads from 2 females) were collected. For E10.5 and E11.5: corresponding tails were labeled for identification and placed in 1x PBS on ice. We performed quick Quinacrine and DAPI staining (~15 min) of corresponding E10.5 and E11.5 fetal tail samples to eliminate the male gonad samples with positive Quinacrine staining of tail samples. We then pooled the dissected female gonad sample and dissociated it into single cells using 0.25% Trypsin at $37^0$ C for 5–7 min with two pipet trituration in between. Fetal bovine serum (10 %) was then used to neutralize trypsin. Dissociated cells were passed through a 100 µm strainer. The cell suspension was centrifuged at 300 g for 5 min and the cell pellet was resuspended in freshly prepared and

filter-sterilized 0.04% BSA. Viability of cells was assessed via trypan staining, and >90 percent viable samples were selected and loaded (~10,000 live cells for E11.5 and E15.5, ~20,000 cells for E10.5) onto the 10 X Genomics Chromium Single Cell system using the for E11.5 and E15.5-v3 and for E10.5 v4 chemistry as per the manufacturer's instruction. Single-cell RNA capture and library preparations were performed, and standard data processing was performed using Cell Ranger pipeline (6.0.1 - for E11.5 and E15.5 and 8.0.1 - for E10.5) and later data was visualized and analyzed by Seurat v5.1.0.

## Single-cell RNA sequencing of *Dazl* mutant gonad

*Dazl*$^{+/-}$ female and male mice were kept for mating, and plugged females were marked as E0.5. The gonad from E11.5 and E12.5 fetuses were dissected, and fetuses were collected in and kept in cold PBS in a 12-well plate such that each well with an individual fetus was marked for identification. Fetal tails were collected, labeled and subjected to Kapa Express Extract Kit (Catalog #KR0383-v4.16) mediated fast DNA extraction according to manufacturer's protocol. Briefly, one-step lysis and DNA extraction system was set up in 100 µl volume by adding 10 µl express extract buffer, 2 µl of express extract enzyme and 88 µl of PCR grade water in each tail sample followed by 75 °C incubation in heating block for 15 min for lysis. The samples were then incubated at 95 °C in a heating block for 5 min for DNA extraction. The sample was then centrifuged briefly to pellet the debris, and supernatant was used directly for PCR as mentioned in the methods section of RNA isolation, cDNA synthesis and PCR. The standard genotyping JAX protocol for strain 035880 is referred to identify homozygous / heterozygous and wild-type fetuses. Simultaneously, male/female sexing was also performed (primer details in Appendix 1-Key resources table). After correct identification, homozygous fetuses from E11.5 (6 gonads from 3 females) and E12.5 (6 gonads from two females) were trypsinized to prepare live single cells, which were then subjected to 10 X Genomics Chromium (v3 chemistry), and data was processed using cell ranger pipeline 6.0.1 and analyzed using seurat v5.1.0 in the same manner as mentioned previously for E11.5 gonad.

## Cell identification and clustering analysis

Single-cell RNA sequencing (scRNA-seq) data were analyzed using the Seurat package (v5.1.0, https://satijalab.org). Count data generated by the Cell Ranger pipeline (6.0.1/8.0.1) were imported into R using the Read10X function and converted into a Seurat object with the CreateSeuratObject function. R packages were used to filter out the low-quality cells, and the following criteria were used to filter cells:

a. **for E10.5 WT gonad**: nFeature_RNA >500 & nFeature_RNA <8500 & nCount_RNA >500 & percent.mt<5.
b. **for E11.5 WT gonad**: nFeature_RNA >100 & nFeature_RNA <11000 & nCount_RNA >300 & percent.mt<10.
c. **for E15.5 WT ovary:** nFeature_RNA >500 & nFeature_RNA <9000 & nCount_RNA >500 & percent.mt<10.

The filter count matrices were Log-normalized using the 'NormalizeData' function and scaled with the 'ScaleData' function to prepare for dimensionality reduction. 'Principal Component Analysis (PCA)' was then applied to reduce dimensionality, followed by using top 15 dimensions and default resolution to cluster cells based on gene expression profiles using the 'FindNeighbors' and 'FindClusters' functions. Cell populations were visualized using the UMAP method, facilitating the identification of distinct cell types.

## Bioinformatic segregation of female germ cells

To segregate E10.5 and E11.5 XX-specific germ cells with complete surety, we performed bioinformatic segregation of female germ cells to address the limitations of primitive Quinacrine staining. The expression of the X-linked gene Xist, Ddx3x, Utx, etc., were used as a marker. Cells with significant Xist or other X-linked gene expression expression were labeled as 'XXonly' using the 'Idents' and 'WhichCells' R functions. A new Seurat object containing only XX-specific cells was created using the subset function. Germ cells within this subset were identified by examining the expression of germ cell-specific marker genes, and a further subset of E10.5 and E11.5 female germ cells was created using the subset function to focus on germ cells specifically.

## Wild-type germ cells-merged dataset creation and validation

ScRNA-seq datasets from E11.5 to P5 (retrieved from the GEO database, GSE136441 and merged object submitted to Github) were visualized using Seurat. E11.5 cells were discarded and a new merged dataset from E10.5-P5 was created, subsets of germ cell clusters from E10.5, E11.5, and E15.5 were integrated using Seurat's 'Merge' function. The resulting merged dataset was then processed by normalizing and scaling the data, followed by the identification of variable genes. The batch correction was performed using pre-harmony and post-harmony data was assessed by visualizing LISI (Local Inverse Simpson's Index) score confirming successful batch mixing. Cell clusters were determined using Seurat's shared nearest neighbor (SNN) algorithm with PCA reduction using top 15 dimensions and 2.8 resolution. To visualize these clusters, dimensionality reduction was applied (umap) with the following criteria: RunUMAP(new_merge, dims = 1:15, n.neighbors=40, min.dist=0.6, spread = 1) enabling the identification of distinct cell populations.

Cell cycle stages of germ cells across the different developmental stages (E10.5-P5) were validated by visualizing meiosis stage-specific expression pattern via feature plot. This visualization technique allows for the capture of subtle trends and helps in comparing expression peaks, providing insights into the transition from mitosis to meiosis across the different developmental stages. Additionally, a violin plot was employed to visualize the variability in average raw UMI levels across germ cells. This approach allowed for an effective comparison of gene expression profiles, highlighting the differences in gene expression patterns between different stages.

### *Dazl* mutant merged dataset

For E11.5 and E12.5 *Dazl*$^{-/-}$ dataset, cells were filtered based on the following criteria:

a. **for E11.5 -/- ovary**: nFeature_RNA >500 & nFeature_RNA <9000 & nCount_RNA >500 & percent.mt<5.
b. **for E12.5 Dazl-/- ovary** nFeature_RNA >500 & nFeature_RNA <10000 & nCount_RNA >500 & percent.mt<5.

The top 15 dimensions were used at default resolution to cluster the cells which were then visualized by UMAP. To create a merged dataset from the subset containing only germ cells were created and merged using the 'merge' function as described previously. Three datasets were created - for E11.5 wild-type and *Dazl*$^{-/-}$ dataset, for E12.5 and *Dazl*-/- dataset and one dataset was created where both E11.5 and 12.5 WT and *Dazl*-/- were merged. Batch correction was applied using harmony. The merged dataset was normalized, scaled, and the top 15 dimensions at a resolution of 0.4 was used for visualization using UMAP.

### In vitro gonad culture: Microtubule inhibitor/BFA treatment and organelle tracker assay

To perform in vitro culture of mouse fetal ovaries on membrane inserts with or without inhibitors, we prepared sterile culture medium, constituting DMEM/F-12 supplemented with 10% fetal bovine serum and penicillin-streptomycin. Inhibitor stock solutions were prepared in DMSO and diluted at desi concentration in culture media (Ciliobrevin D from Millipore, 25 μM for 6 hr, BFA from Invitrogen, 5 μg/ml for 6 hr). A control medium containing the vehicle DMSO was prepared simultaneously. Fetal ovaries were dissected from E11.5 embryos under a stereomicroscope in sterile PBS. We have kept the surrounding mesonephric tissue intact and attached to the gonad to ensure proper development. We have also collected the corresponding fetal tail of each fetal gonad to perform genetic sex determination. We have placed membrane inserts into a 24-well plate and added ~500 μL of culture medium (with or without inhibitors) to the lower chamber and ensured no contact between medium and the insert surface. Next, we gently placed isolated ovaries onto the membrane surface using sterile forceps. The plate is then incubated in a $CO_2$ incubator at 37 °C for ~6 hr. After the designated culture period, confirmed ovaries (post genetic-sex analysis) were carefully removed from the membrane inserts and washed three times with basal media to remove any residual inhibitors. Next, we collected ovaries and fixed them in 4% PFA for downstream immunolocalization studies. To inhibit the microtubule growth using cold treatment, we transfer dissected gonad to a chilled culture medium, maintained at 0–4°C. The tissue is fully immersed in the pre-cooled medium and incubated on ice for ~60 min. This low-temperature exposure destabilizes microtubules by disrupting tubulin

dynamics, effectively depolymerizing existing microtubules and inhibiting further polymerization. After the incubation, the tissue is washed with fresh ice-cold medium and fixed in 4% PFA. To stain specifically for organelles: Fetal ovaries were dissected and cultured in vitro in DMEM/F12 medium supplemented with organelle-specific fluorescent dyes. LysoTracker Deep (Thermo Fisher, L7528, 1 mM stock) or MitoTracker (Thermo Fisher, M7514) or MitoTracker Deep (Thermo Fisher, M22426), and ER-Tracker (Thermo Fisher, E34251) or ER-Tracker (Thermo Fisher, E34250) were each added at a final concentration according to the manufacturer's recommendations. The ovaries were incubated in this dye-containing medium at 37 °C with 5% $CO_2$ for 12 hr. Following incubation, tissues were collected and fixed in 4% paraformaldehyde (PFA) for subsequent immunostaining.

## Magnetic activated cell sorting (MACS)

Embryonic ovaries were pooled and treated with trypsin, then were neutralized followed by centrifugation as mentioned before in the scRNA-sequencing method section to create dissociated single cells. The resulting cell pellet was re-suspended in ~80 microliters MACS buffer i.e., PBS with 0.5% BSA and 2 mM ethylene diaminetetraacetic acid (EDTA). ~20 microliters Anti-SSEA1 (CD15) microbeads (Miltenyi Biotec Inc, #130-094-530) were added to the cell suspension followed by incubation for 20 min on ice. After adding 1 ml MACS Buffer to the suspension, the SSEA1 bound cells were pelleted by centrifugation at 300 g for 10 min at 4 °C. SSEA1+ve and SSEA1-ve cells were then separated by applying cell suspension to MS columns according to the manufacturer's instruction (Miltenyi Biotec Inc). The SSEA1+ve cells were retained by the column and were filtered thrice to obtain a pure germ cell population. The cells were counted, and viability was assessed via trypan staining and using an automated cell counter (Fisher Scientific). For validation of the MACS protocol, the cells were fixed in 4% PFA to perform immunostaining of DDX4 and were also stored in Trizol to perform RNA synthesis and PCR for germ cell-specific gene expression.

## IRE1-Xbp1 assay

The IRE1-XBP1 ratiometric assay (from Montana Molecular #U0921G) is a genetically encoded biosensor system that utilizes a BacMam Vector carrying dual-fluorescence biosensor. For each assay reaction, 50 µl transduction mix consisting of XBP1-IRE1 sensor (15 µl) plus sodium butyrate (0.6 µl per well) was prepared in basal media. To standardize the working conditions, manufacturer-provided thapsigargin (1 µm per well) was used as positive control, and untransduced cells were used as negative control. For each experiment, equal amounts of SSEA1-ve and SSEA1+ve cells were added to a 96-well plate in duplicates. ~50 microliters of transduction mix was added to each reaction well. The plate was incubated for 45 min at room temperature and then transferred to 5% CO2 and 37 °C for 24 hr. The enzymatic activity was then measured at 37 °C by monitoring XBP1-IRE-1 sensor's fluorescent intensity (top read) at Ex/Em = 488/525 nm and basal constitutive fluorescent intensity at 565/620 using microplate reader (BioTek Synergy H1). IRE1-Xbp1 assay comparing MACS sorted SSEA1$^+$ germ vs SSEA1$^-$ somatic cells at E11.5 was performed across six experiments (~32 mice, ≥5 mice, and ≥20 ovaries per experiment). IRE1-Xbp1 assay comparing SSEA1$^+$ vs SSEA1$^-$ E12.5 cells in WT or *Dazl* mutant mouse across 3 experiments (~40 mice, ≥5 mice, and ≥25 ovaries per experiments).

## 20S proteasome activity assay

To perform proteasome activity assay (Amplite #13456) we followed manufacturer's instructions. We first standardized the working conditions using trypsin enzyme as a technical positive control. Trypsin is loaded on a different well in different concentrations in duplicates. Blank medium along with assay buffer were added in separate wells to act as a negative control. For each experiment, equal amounts of SSEA1-ve and SSEA1+ve cells were added to a 96-well plate in duplicates. ~50 microliters of working solution consisting of fluorogenic substrate in assay buffer (prepared according to manufacturer's instructions) were added and the plate was incubated for 2 hr at 37 °C. The enzymatic activity was then measured at 37 °C by monitoring fluorescent intensity (top read) at Ex/Em = 490/525 nm (Cut off = 515 nm) using microplate reader (BioTek Synergy H1). Proteasome activity in MACS-sorted SSEA1$^+$ vs SSEA1$^-$ cells were assessed using three biological assays with ~35–60 E11.5 ovaries per assay. Proteasome activity ratio of MACS-sorted SSEA1$^+$ and SSEA1$^-$ cells compared at E12.5 with ~25–28 E12.5 ovaries were used per assay.

### RNA isolation and cDNA synthesis

Total RNA was extracted using TRIzol reagent (Invitrogen) following the manufacturer's instructions. Briefly, tissue samples were homogenized in TRIzol, or cells were incubated in TRIzol for 5 min at RT, followed by phase separation with chloroform (200 μl/ml TRIzol, shake for 10–15 s and keep at RT for 5 min) and centrifugation at 12,000×g for 15 min at 4 °C. The aqueous phase was collected, and RNA was precipitated with an equal amount of isopropanol (10 min RT then same centrifugation as last step), washed with 75% ethanol, centrifuged, removed the supernatant ethanol and air-dried. Resuspended the pellet in RNase-free water. RNA quality and concentration were determined spectrophotometrically. For cDNA synthesis, we used the Superscript IV Cells Direct cDNA synthesis kit (Invitrogen, #11750150) with a modified manufacturer's protocol. We skipped the lysis step and directly added DNAse I to our sample (0.5 μl each) followed by 10 min incubation on ice. Then we added Stop solution (3 μl each) and kept the mixture at RT for 2 min. Then we set up the RT reaction as per manufacturer's instruction by adding 8 μl of RT mix to each sample and setting up the following reaction in PCR thermocycler: 25 °C for 10 min, 55 °C for 10 min, 85 °C for 5 min and hold at 4°C. cDNA was used directly for PCR as mentioned in Sexing and genotyping using DDX4 and GAPDH Primers (See Primer details).

## Quantification and statistical analysis

### Quantification of germ cell numbers

PFA-fixed ovaries were subjected to whole mount immunostaining as described previously. Z-stack images for whole ovaries were analyzed using FIJI. Germ cell numbers are quantified in each ovary by manually counting EMA/DDX4 stained germ cells in multiple 0.45 μm optical sections from different ovaries using FIJI.

### 3D volume quantification

Whole mount immunostaining was performed and the z-stack images were converted to 3D image to analyze using Imaris as described previously. After using the Surfaces module to create surface rendering and applying manual thresholding and smoothing to define the structure accurately, we navigated to the statistics tab to access volume measurements. Further visualization of the results was performed in 3D or orthogonal views to confirm accuracy and adjusted rendering properties, such as color or transparency as needed. On average, the average values from three measurements performed were considered and plotted as a graph using GraphPad Prism.

### Area of staining quantified using FIJI

Images were analyzed using FIJI to calculate area of staining by selecting region of interest and using measure command from Analyze tab. The results were plotted as the number of pixels to depict the area corresponding to stained regions.

### Statistics

Data are presented as mean ± SEM (standard error of the mean). Statistical analyses were performed using Student's paired t-test, one-way ANOVA with Tukey's post-hoc test, or the Friedman test with Dunn's post-hoc test for non-parametric data. Significance is indicated as follows: $p<0.05$ (*), $p<0.01$ (**), $p<0.005$ (***), and $p<0.001$ (****). Each experiment was performed on 'N' individual specimens, as indicated in the figure legend, from at least three different animals.

## Resource availability

Lead contact: Further information and requests for resources and reagents should be directed to and will be fulfilled by the Lead Contact, Allan C. Spradling (spradling@carnegiescience.edu).

### Materials availability

This study did not generate new unique reagents.

## Acknowledgements

We thank Wanbao Niu and Mike Sepanski for generating a laboratory archive of EM images of mouse ovaries that was used to prepare *Figures 1E and 5H*. We thank Wanbao Niu, Qi Yin, and Ashish Tiwari for sharing insights on mouse developmental genetic technology and its application to the ovary. The authors thank Ru-ching Hsia for assistance in electron microscopy including the images of *Figure 1E'*, *Figure 1—figure supplement 1K*. We thank Dr. Eugenia Dikovsky for skillfully managing the Carnegie mouse facility. We thank Allison Pinder and Dr. Feric Tan for assistance with genomics. We thank members of the Spradling lab for helpful comments throughout the course of the research and publication process. Allan Spradling is a staff member of the Carnegie Institution for Science who hosts the laboratory at its former Department of Embryology and provides generous additional support.

## Additional information

### Funding

| Funder | Grant reference number | Author |
| --- | --- | --- |
| Howard Hughes Medical Institute | | Allan C Spradling |

The funders had no role in study design, data collection and interpretation, or the decision to submit the work for publication.

### Author contributions

Madhulika Pathak, Conceptualization, Data curation, Software, Formal analysis, Validation, Investigation, Visualization, Methodology, Writing – original draft, Writing – review and editing; Allan C Spradling, Conceptualization, Resources, Data curation, Software, Formal analysis, Supervision, Funding acquisition, Validation, Visualization, Methodology, Writing – original draft, Project administration, Writing – review and editing

### Author ORCIDs

Madhulika Pathak ⓘ https://orcid.org/0009-0009-3040-531X
Allan C Spradling ⓘ https://orcid.org/0000-0002-5251-1801

### Ethics

This study was performed in accordance with the Guide for the Care and Use of Laboratory Animals of the National Institutes of Health. All of the animals were handled according to approved animal care and use committee (IACUC) protocols (#126) of the Carnegie Institution. The protocol was last re-approved on 4/10/2025 by the Carnegie Institution Animal Care and Use Committee (Animal Welfare Assurance Number A3861-01).

Reviewer #2 (Public review): https://doi.org/10.7554/eLife.109358.3.sa1
Reviewer #3 (Public review): https://doi.org/10.7554/eLife.109358.3.sa2
Author response https://doi.org/10.7554/eLife.109358.3.sa3

## Additional files

### Supplementary files

MDAR checklist

### Data availability

The single-cell RNA sequencing of wild-type and/or *Dazl* mutant mouse ovarian cells across embryonic developmental stages E10.5 (Wild type), E11.5 (Wild type and *Dazl-/-*), E12.5 (*Dazl-/-*), E15.5 (Wild type) have been deposited and are available publicly in the NIH Gene Expression Omnibus (GEO) database under accession number GSE303512. (https://www.ncbi.nlm.nih.gov/geo/query/acc.cgi?acc=GSE303512).

The following dataset was generated:

| Author(s) | Year | Dataset title | Dataset URL | Database and Identifier |
|---|---|---|---|---|
| Pathak M, Spradling AC | 2026 | Single-cell RNA Sequencing of Wildtype and/or Dazl Mutant Mouse Ovarian Cells Across Embryonic Developmental Stages E10.5 (Wild type), E11.5 (Wild type and Dazl-/-), E12.5 (Dazl-/-), E15.5 (Wild type) | http://www.ncbi.nlm.nih.gov/geo/query/acc.cgi?acc=GSE303512 | NCBI Gene Expression Omnibus, GSE303512 |

The following previously published dataset was used:

| Author(s) | Year | Dataset title | Dataset URL | Database and Identifier |
|---|---|---|---|---|
| Niu W, Spradling AC | 2020 | Single-cell analysis RNA sequencing of prenatal and neonatal gonads/ovaries at E11.5, E12.5, E14.5, E16.5, E18.5, P1 and P5 | https://www.ncbi.nlm.nih.gov/geo/query/acc.cgi?acc=GSE136441 | NCBI Gene Expression Omnibus, GSE136441 |

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

# Appendix 1

## Appendix 1—key resources table

| Reagent type (species) or resource | Designation | Source or reference | Identifiers | Additional information |
|---|---|---|---|---|
| Gene (*Mus musculus*) | Dazl | NCBI | https://www.ncbi.nlm.nih.gov/Dazl | |
| Gene (*Mus musculus*) | Pard3 | NCBI | https://www.ncbi.nlm.nih.gov/Pard3 | |
| Gene (*Mus musculus*) | Xbp1 | NCBI | https://www.ncbi.nlm.nih.gov/Xbp1 | |
| Strain, strain background (*Mus musculus*) | C57BL/6 J | Jackson Laboratory | 000664; RRID:IMSR_JAX:000664 | |
| Genetic reagent (*Mus musculus*) | CAG-Cre-ER | Jackson Laboratory | 004682, RRID:IMSR_JAX:004682 | |
| Genetic reagent (*Mus musculus*) | *Dazl*-1L -/+ | Jackson Laboratory | RRID:IMSR_JAX:035880 | |
| Genetic reagent (*Mus musculus*) | R26R-EYFP | Jackson Laboratory | 006148; RRID:IMSR_JAX:006148 | |
| Biological sample (*Mus musculus*) | fetal ovary | | | |
| Antibody | EMA | DSHB | EMA-1 RRID:AB_531885 | IF (1:1) Culture supernatant |
| Antibody | GCNA | Abcam | ab82527 RRID:AB_1659152 | IF (1:400) |
| Antibody | DDX4 | Abcam/R&D systems | ab13840 RRID:AB_443012/ab27591 RRID:AB_11139638/AF2030 RRID:AB_2277369 | IF (1:400) |
| Antibody | SSEA1 | DSHB | MC-480 RRID:AB_528475 | IF (1:1) Culture supernatant |
| Antibody | Gm-130 | BD Biosciences/ Novus Biologicals | 610822, RRID:AB_398141/ NBP2-53420, RRID:AB_2916095 | IF (1:200) |
| Antibody | Rab9 | Thermo Fisher | MA5-31997, RRID:AB_2809291 | IF (1:200) |
| Antibody | Tex14 | Proteintech | 18351–1-AP, RRID:AB_10641992 | IF (1:400) |
| Antibody | GFP/YFP | Aves Labs | GFP-1020, RRID:AB_10000240 | IF (1:600) |
| Antibody | Dazl | BioRad Labs/ GeneTex | MCA2336, RRID:AB_2292585/ GTX89448, RRID:AB_10722773 | IF (1:100) |
| Antibody | Lysotracker deep | Thermo Fisher | L7528 | IF (1:600) |
| Antibody | Pard3 | Novus Biologicals | NBP1-88861, RRID:AB_11056253 | IF (1:200) |
| Antibody | LAMP1 | Cell Signaling Technology | 9091, RRID:AB_2687579 | IF (1:200) |
| Antibody | Xbp1 | Abcam | ab37152, RRID:AB_778939 | IF (1:200) |
| Antibody | Gs28 | BD Biosciences | 61184, RRID:AB_398718 | IF (1:200) |
| Antibody | Sec63 | Thermo Fisher | PA5-100180, RRID:AB_2815710 | IF (1:200) |
| Antibody | Calnexin | Abcam | ab219644, RRID:AB_3732991 | IF (1:200) |
| Antibody | Gs28 | Proteintech | CL555-16106, RRID:AB_2919629 | IF (1:200) |
| Antibody | Acetyl-α-Tubulin (Lys40) | Cell Signaling. Tech | 5335, RRID:AB_10544694 | IF (1:600) |
| Antibody | Alpha Tubulin (acetyl K40) | Abcam | ab289875, AB_3733017 | IF (1:100) |
| Antibody | Pericentrin | Abcam | ab4448, RRID:AB_304461 ab28144, RRID:AB_2160664 | IF (1:200) |
| Antibody | Rac GAP1 Antibody (A-6) | Santa Cruz Biotechnology, Inc | sc-271110, RRID:AB_10611939 | IF (1:200) |

*Appendix 1 Continued on next page*

*Appendix 1 Continued*

| Reagent type (species) or resource | Designation | Source or reference | Identifiers | Additional information |
|---|---|---|---|---|
| Antibody | Dnmt3a | Cell Signaling. Tech | 3598, RRID:AB_2277449 | IF (1:200) |
| Antibody | ATP5A | Abcam | ab14748, RRID:AB_301447 | IF (1:200) |
| Sequence-based reagent | *Uba1*_Forward | PCR Primers | *McFarlane et al., 2013* | 5´-TGGTCTGGACCCAAAC GCTGTCCACA-3´ |
| Sequence-based reagent | *Uba1*_Reverse | PCR Primers | *McFarlane et al., 2013* | 5´-GGCAGCAGCCATCACATAAT CCAGATG-3´, |
| Sequence-based reagent | *Sly*_Forward | PCR Primers | *McFarlane et al., 2013* | 5'-GATGATTTGAGTGGAAATGT GAGGTA-3' |
| Sequence-based reagent | *Sly*_Reverse | PCR Primers | *McFarlane et al., 2013* | 5'-CTTATGTTTATAGGCATGCA CCATGTA-3' |
| Sequence-based reagent | *Zfy*_Forward | PCR Primers | *McFarlane et al., 2013* | 5'-GACTAGACATGTCTTAACAT CTGTCC-3' |
| Sequence-based reagent | *Zfy*_Reverse | PCR Primers | *McFarlane et al., 2013* | 5'-CCTATTGCATGGACTGCAGC TTATG-3' |
| Sequence-based reagent | *Ddx4* Forward | PCR Primers | *Gao et al., 2011* | 5'-GAGATTGCCTTCAGTACCTA TGTG-3' |
| Sequence-based reagent | *Ddx4* Reverse | PCR Primers | *Gao et al., 2011* | 5'-GTGCTTGCCCTGGTAATTCT -3' |
| Sequence-based reagent | *Gapdh* Forward | PCR Primers | *Wang et al., 2011* | 5'-GGTGAAGCAGGCATCT GAGGG-3' |
| Sequence-based reagent | *Gapdh* Reverse | PCR Primers | *Wang et al., 2011* | 5'-GGTGGGTGGTCCAGGGTT-3' |
| Sequence-based reagent | *Dazl* Common | PCR Primers | *JAX Protocol (Strain #035880, Primer# 56340)* | 5'-GAC ATT ACT AAG AAA ACA GCA GTG G-3' |
| Sequence-based reagent | *Dazl* WT reverse | PCR Primers | *JAX Protocol (Primer# 56341)* | 5'-TTC TGC ACA TCC ACG TCA TT-3' |
| Sequence-based reagent | *Dazl* Mut Reverse | PCR Primers | *JAX Protocol (Primer# 56342)* | 5'-ATC CCT CCC TTT AGG GCT CA-3' |
| Chemical compound, drug | AAL | Vector Labs Inc | FL-1391–1 | IF (1:200) |
| Chemical compound, drug | LCA | Vector Labs Inc | FL-1041–5 | IF (1:200) |
| Chemical compound, drug | WGA | Thermo Fisher | W7024 | IF (1:1000) |
| Chemical compound, drug | Paraformaldehyde | Electron Microscopy Sci. | 15714 | Concentration 4% |
| Chemical compound, drug | Corn Oil | Sigma Chemical | C8267 | |
| Chemical compound, drug | Tween 20 | Sigma Chemical | P1379 | |
| Chemical compound, drug | Triton X | Sigma Chemical | X100 | |
| Chemical compound, drug | Mounting media | Vector Labs Inc. | H-1000 | |
| Chemical compound, drug | Trypsin-EDTA (0.25%) | Fisher | 25200056 | |
| Chemical compound, drug | Fetal Bovine serum | Sigma Chemical | F2442 | chemical compound, drug |
| Chemical compound, drug | Bovine serum albumin | Sigma Chemical | A4503 | chemical compound, drug |
| Chemical compound, drug | SSEA1 (CD15) microbeads | Miltenyi Biotech | 130-094-530 | chemical compound, drug |

*Appendix 1 Continued on next page*

*Appendix 1 Continued*

| Reagent type (species) or resource | Designation | Source or reference | Identifiers | Additional information |
|---|---|---|---|---|
| Chemical compound, drug | DMEM/F-12 | Fischer | 11320–033 | chemical compound, drug |
| Chemical compound, drug | Celiobrevin D | Millipore | 250401 | chemical compound, drug |
| Chemical compound, drug | Tamoxifen | Sigma Chemical | T5648 | |
| Commercial assay or kit | Mitotracker /Deep | Thermo Fischer | M7514/M22426 | IF (1:10000) |
| Commercial assay or kit | ER-Tracker / | Thermo Fischer | E34251/E34250 | IF (1:10000) |
| Commercial assay or kit | Kappa fast hot start ready-mix | KAPPA Biosystems | KK5608 | |
| Commercial assay or kit | KAPPA express extract | KAPPA Biosystems | KR0383-v4.16 | |
| Commercial assay or kits | IRE1-Xbp1 Assay | Montana Molecular | U0921G | |
| Commercial assay or kit | Proteasome Activity | Amplite | 13456 | |
| Commercial assay or kit | Superscript IV cells direct cDNA synthesis kit | Fischer | 11750510 | |
| Software, algorithm | IMARIS | Oxford Instruments (formerly Bitplane) | Version: 10.2, RRID:SCR_007370 | IMARIS |
| Software, algorithm | SEURAT | Satija Lab | Version: 5.1.0, RRID:SCR_016341 | SEURAT |
| Software, algorithm | CELL RANGER PIPELINE | 10 X Genomics | Version: 6.0.1/8.0.1, RRID:SCR_017344 | CELL RANGER PIPELINE |
| Software, algorithm | GRAPHPAD PRISM | GraphPad Software | Version: 10.5.0, RRID:SCR_002798 | GRAPHPAD PRISM |
| Other | MACS MS separation columns | Miltenyi Biotech | 130-042-201 | MACS MS separation columns |
| Other | Cell culture inserts | Millipore | PICM01250 | Cell culture inserts |
| Other | Cell strainer | Sigma | CLS431752 | Cell strainer |

