## [Editor Report · eLife Assessment]

This manuscript provides evidence that mouse germline cysts develop an asymmetric Golgi, ER, and microtubule-associated structure that resembles the fusome in Drosophila germline cysts. This **fundamental** study provides new evidence that fusome-like structures exist in germ cell cysts across species. Overall, the data are **convincing** and represent a significant advance in our understanding of germ cell biology.

---

## [Referee Report · Reviewer #2 (Public review)]

This study identifies Visham, an asymmetric structure in developing mouse cysts resembling the Drosophila fusome, an organelle crucial for oocyte determination. Using immunofluorescence, electron microscopy, 3D reconstruction, and lineage labeling, the authors show that primordial germ cells (PGCs) and cysts, but not somatic cells, contain an EMA-rich, branching structure that they named Visham, which remains unbranched in male cysts. Visham accumulates in regions enriched in intercellular bridges, forming clusters reminiscent of fusome "rosettes." It is enriched in Golgi and endosomal vesicles and partially overlaps with the ER. During cell division, Visham localizes near centrosomes in interphase and early metaphase, disperses during metaphase, and reassembles at spindle poles during telophase before becoming asymmetric. Microtubule depolymerization disrupts its formation.

Cyst fragmentation is shown to be non-random, correlating with microtubule gaps. The authors propose that 8-cell (or larger) cysts fragment into 6-cell and 2-cell cysts. Analysis of Pard3 (the mouse ortholog of Par3/Baz) reveals its colocalization with Visham during cyst asymmetry, suggesting that mammalian oocyte polarization depends on a conserved system involving Par genes, cyst formation, and a fusome-like structure.

Transcriptomic profiling identifies genes linked to pluripotency and the unfolded protein response (UPR) during cyst formation and meiosis, supported by protein-level reporters monitoring Xbp1 splicing and 20S proteasome activity. Visham persists in meiotic germ cells at stage E17.5 and is later transferred to the oocyte at E18.5 along with mitochondria and Golgi vesicles, implicating it in organelle rejuvenation. In Dazl mutants, cysts form, but Visham dynamics, polarity, rejuvenation, and oocyte production are disrupted, highlighting its potential role in germ cell development.

Overall, this is an interesting and comprehensive study of a conserved structure in the germline cells of both invertebrate and vertebrate species. Investigating these early stages of germ cell development in mice is particularly challenging. Although primarily descriptive, the study represents a remarkable technical achievement. The images are generally convincing, with only a few exceptions.

Major comments:

(1) Some titles contain strong terms that do not fully match the conclusions of the corresponding sections.

(1a) Article title "Mouse germline cysts contain a fusome-like structure that mediates oocyte development":

The term "mediates" could be misleading, as the functional data on Visham (based on comparing its absence to wild-type) actually reflects either a microtubule defect or a Dazl mutant context. There is no specific loss-of-function of visham only.

(1b) Result title, "Visham overlaps centrosomes and moves on microtubules":

The term "moves" implies dynamic behavior, which would require live imaging data that are not described in the article.

(1c) Result title, "Visham associates with Golgi genes involved in UPR beginning at the onset of cyst formation":

The presented data show that the presence of Visham in the cyst coincides temporally with the expression and activity of the UPR response; the term "associates" is unclear in this context.

(1d) Result title, "Visham participates in organelle rejuvenation during meiosis":

The term "participates" suggests that Visham is required for this process, whereas the conclusion is actually drawn from the Dazl mutant context, not a specific loss-of-function of visham only.

(2) The authors aim to demonstrate that Visham is a fusome-like structure. I would suggest simply referring to it as a "fusome-like structure" rather than introducing a new term, which may confuse readers and does not necessarily help the authors' goal of showing the conservation of this structure in Drosophila and Xenopus germ cells. Interestingly, in a preprint from the same laboratory describing a similar structure in Xenopus germ cells, the authors refer to it as a "fusome-like structure (FLS)" (Davidian and Spradling, BioRxiv, 2025).

Comments on revisions:

The revised manuscript has been clearly improved, and the authors have addressed all of our comments. I would like to point out two minor issues:

(1) As suggested by the reviewers, the authors now use the term fusome instead of visham. However, they also acknowledge that this structure lacks many components of the Drosophila fusome. It may therefore be more appropriate to refer to it as a "mouse fusome" or as a "fusome-like structure (FLS)," as used in Xenopus.

(2) I agree with Reviewer 3 that co-localization between EMA and acTubulin on still images does not convincingly demonstrate that fusome vesicles move along microtubules (Figure S2E).

---

## [Referee Report · Reviewer #3 (Public review)]

The manuscript provides evidence that mice have a fusome, a conserved structure most well studied in Drosophila that is important for oocyte specification. Overall, a myriad of evidence is presented demonstrating the existence of a mouse fusome. This work is important as it addresses a long-standing question in the field of whether mice have fusomes and sheds light on how oocytes are specified in mammals.

Comments on revisions:

Overall, the authors did a good job of responding to reviewer comments that have improved the manuscript by including higher quality microscope images, revising text for clarity and using the term mouse fusome instead of using a new term. However, two of the headings in the results section that didn't correspond to the data presented in that section still have not been revised eventhough the authors stated that they were revised in their response to reviewer comments. The heading of the first section of the results is: "PGCs contain a Golgi-rich structure known as the EMA granule" even though no evidence in that section shows it is Golgi rich. The heading of the fifth section of the results is: "The mouse fusome associates with polarity and microtubule genes including pard3" however, only evidence for pard3 is presented.

---

## [Author Response]

The following is the authors’ response to the original reviews.

**Public Reviews:**

**Reviewer #1 (Public review)**
Summary

We thank the reviewer for the constructive and thoughtful evaluation of our work. We appreciate the recognition of the novelty and potential implications of our findings regarding UPR activation and proteasome activity in germ cells.

(1) The microscopy images look saturated, for example, Figure 1a, b, etc. Is this a normal way to present fluorescent microscopy?

The apparent saturation was not present in the original images, but likely arose from image compression during PDF generation. While the EMA granule was still apparent, in the revised submission, we will provide high-resolution TIFF files to ensure accurate representation of fluorescence intensity and will carefully optimize image display settings to avoid any saturation artifacts.

(2) The authors should ensure that all claims regarding enrichment/lower vs. lower values have indicated statistical tests.

We fully agree. In the revised version, we will correct any quantitative comparisons where statistical tests were not already indicated, with a clear statement of the statistical tests used, including p-values in figure legends and text.

(a) In Figure 2f, the authors should indicate which comparison is made for this test. Is it comparing 2 vs. 6 cyst numbers?

We acknowledge that the description was not sufficiently detailed. Indeed, the test was not between 2 vs 6 cyst numbers, but between all possible ways 8-cell cysts or the larger cysts studied could fragment randomly into two pieces, and produce by chance 6-cell cysts in 13 of 15 observed examples. We will expand the legend and main text to clarify that a binomial test was used to determine that the proportion of cysts producing 6-cell fragments differed very significantly from chance.

Revised text:

“A binomial test was used to assess whether the observed frequency of 6-cell cyst products differed from random cyst breakage. Production of 6-cell cysts was strongly preferred (13/15 cysts; ****p < 0.0001).”

(b) Figures 4d and 4e do not have a statistical test indicated.

We will include the specific statistical test used and report the corresponding p-values directly in the figure legends.

(3) Because the system is developmentally dynamic, the major conclusions of the work are somewhat unclear. Could the authors be more explicit about these and enumerate them more clearly in the abstract?

We will revise the abstract to better clarify the findings of this study. We will also replace the term Visham with mouse fusome to reflect its functional and structural analogy to the Drosophila and Xenopus fusomes, making the narrative more coherent and conclusive.

(4) The references for specific prior literature are mostly missing (lines 184-195, for example).

We appreciate this observation of a problem that occurred inadvertently when shortening an earlier version. We will add 3–4 relevant references to appropriately support this section.

(5) The authors should define all acronyms when they are first used in the text (UPR, EGAD, etc).

We will ensure that all acronyms are spelled out at first mention (e.g., Unfolded Protein Response (UPR), Endosome and Golgi-Associated Degradation (EGAD)).

(6) The jumping between topics (EMA, into microtubule fragmentation, polarization proteins, UPR/ERAD/EGAD, GCNA, ER, balbiani body, etc) makes the narrative of the paper very difficult to follow.

We are not jumping between topics, but following a narrative relevant to the central question of whether female mouse germ cells develop using a fusome. EMA, microtubule fragmentation, polarization proteins, ER, and balbiani body are all topics with a known connection to fusomes. This is explained in the general introduction and in relevant subsections. We appreciate this feedback that further explanations of these connections would be helpful. In the revised manuscript, use of the unified term mouse fusome will also help connect the narrative across sections. UPR/ERAD/EGAD are processes that have been studied in repair and maintenance of somatic cells and in yeast meiosis. We show that the major regulator XbpI is found in the fusome, and that the fusome and these rejuvenation pathway genes are expressed and maintained throughout oogenesis, rather than only during limited late stages as suggested in previous literature.

(7) The heading title "Visham participates in organelle rejuvenation during meiosis" in line 241 is speculative and/or not supported. Drawing upon the extensive, highly rigorous Drosophila literature, it is safe to extrapolate, but the claim about regeneration is not adequately supported.

We believe this statement is accurate given the broad scope of the term "participates." It is supported by localization of the UPR regulator XbpI to the fusome. XbpI is the ortholog of HacI a key gene mediating UPR-mediated rejuvenation during yeast meiosis. We also showed that rejuvenation pathway genes are expressed throughout most of meiosis (not previously known) and expanded cytological evidence of stage-specific organelle rejuvenation later in meiosis, such as mitochondrial-ER docking, in regions enriched in fusome antigens. However, we recognize the current limitations of this evidence in the mouse, and want to appropriately convey this, without going to what we believe would be an unjustified extreme of saying there is no evidence.

**Reviewer #2 (Public review):**

We thank the reviewer for the comprehensive summary and for highlighting both the technical achievement and biological relevance of our study. We greatly appreciate the thoughtful suggestions that have helped us refine our presentation and terminology.

(1) Some titles contain strong terms that do not fully match the conclusions of the corresponding sections.(1a) Article title “Mouse germline cysts contain a fusome-like structure that mediates oocyte development”

We will change the statement to: “Mouse germline cysts contain a fusome that supports germline cyst polarity and rejuvenation.”

(1b) Result title “Visham overlaps centrosomes and moves on microtubules”

We acknowledge that “moves” implies dynamics. We will include additional supplementary images showing small vesicular components of the mouse fusome on spindle-derived microtubule tracks.

(1c) Result title “Visham associates with Golgi genes involved in UPR beginning at the onset of cyst formation”

We will revise this title to: “The mouse fusome associates with the UPR regulatory protein Xbp1 beginning at the onset of cyst formation” to reflect the specific UPR protein that was immunolocalized.

(1d) Result title “Visham participates in organelle rejuvenation during meiosis”

We will revise this to: “The mouse fusome persists during organelle rejuvenation in meiosis.”

(2) The authors aim to demonstrate that Visham is a fusome-like structure. I would suggest simply referring to it as a "fusome-like structure" rather than introducing a new term, which may confuse readers and does not necessarily help the authors' goal of showing the conservation of this structure in Drosophila and Xenopus germ cells. Interestingly, in a preprint from the same laboratory describing a similar structure in Xenopus germ cells, the authors refer to it as a "fusome-like structure (FLS)" (Davidian and Spradling, BioRxiv, 2025).

We appreciate the reviewer’s insightful comment. To maintain conceptual clarity and align with existing literature, we will refer to the structure as the mouse fusome throughout the manuscript, avoiding introduction of a new term.

**Reviewer #3 (Public review):**

We thank the reviewer for emphasizing the importance of our study and for providing constructive feedback that will help us clarify and strengthen our conclusions.

(1) Line 86 - the heading for this section is "PGCs contain a Golgi-rich structure known as the EMA granule"

We agree that the enrichment of Golgi within the EMA PGCs was not shown until the next section. We will revise this heading to:

“PGCs contain an asymmetric EMA granule.”

(2) Line 105-106, how do we know if what's seen by EM corresponds to the EMA1 granule?

We will clarify that this identification is based on co-localization with Golgi markers (GM130 and GS28) and response to Brefeldin A treatment, which will be included as supplementary data. These findings support that the mouse fusome is Golgi-derived and can therefore be visualized by EM. The Golgi regions in E13.5 cyst cells move close together and associate with ring canals as visualized by EM (Figure 1E), the same as the mouse fusomes identified by EMA.

(3) Line 106-107-states "Visham co-stained with the Golgi protein Gm130 and the recycling endosomal protein Rab11a1". This is not convincing as there is only one example of each image, and both appear to be distorted.

Space is at a premium in these figures, but we have no limitation on data documenting this absolutely clear co-localization. We will replace the existing images with high-resolution, noncompressed versions for the final figures to clearly illustrate the co-staining patterns for GM130 and Rab11a1.

(4) Line 132-133---while visham formation is disrupted when microtubules are disrupted, I am not convinced that visham moves on microtubules as stated in the heading of this section.

We will include additional supplementary data showing small mouse fusome vesicles aligned along microtubules.

(5) Line 156 - the heading for this section states that Visham associates with polarity and microtubule genes, including pard3, but only evidence for pard3 is presented.

We agree and will revise the heading to: “Mouse fusome associates with the polarity protein Pard3.” We are adding data showing association of small fusome vesicles on microtubules.

(6) Lines 196-210 - it's strange to say that UPR genes depend on DAZ, as they are upregulated in the mutants. I think there are important observations here, but it's unclear what is being concluded.

UPR genes are not upregulated in DAZ in the sense we have never documented them increasing. We show that UPR genes during this time behave like pleuripotency genes and normally decline, but in DAZ mutants their decline is slowed. We will rephrase the paragraph to clarify that Dazl mutation partially decouples developmental processes that are normally linked, which alters UPR gene expression relative to cyst development.

(7) Line 257-259-wave 1 and 2 follicles need to be explained in the introduction, and how these fits with the observations here clarified.

Follicle waves are too small a focus of the current study to explain in the introduction, but we will request readers to refer to the cited relevant literature (Yin and Spradling, 2025) for further details.

We sincerely thank all reviewers for their insightful and constructive feedback. We believe that the planned revisions—particularly the refined terminology, improved image quality, clarified statistics, and restructured abstract—will substantially strengthen the manuscript and enhance clarity for readers.

**Reviewer #1 (Recommendations for the authors):**
(1) Figure 1E: need to use some immuno-gold staining to identify the Visham. Just circling an area of cytoplasm that contains ER between germ cell pairs is not enough.

We appreciate the reviewer’s insistence that the association between the mouse fusome and Golgi be clearly demonstrated. However, the EMA granule is a large structure discovered and defined by light microscopy, and presents no inherent challenge to documenting its Golgi association by immunofluorescence experiments, which we presented and now further strengthened as described in the next paragraph. We believe that the suggested EM experiment would add little to the EM we already presented (Figure 1E, E') Moreover, due to facility limitations, we are currently unable to perform immunogold staining.

To strengthen previous immunolocalization experiments, we have now included additional immunostaining data showing the clear colocalization of the fusome region with the Golgi markers GM130 and GS28 (Figure S1H). We have also incorporated a new experiment using the Golgi-specific inhibitor Brefeldin A (BFA) see Figure S1I. Treatment of in vitro–cultured gonads with BFA, disrupted EMA granule formation, demonstrating that EMA granules not only associate with Golgi, but require Golgi function to to be maintained.

Additionally, in Figure 2, we showed that the fusome overlaps with the peri-centriolar region—a characteristic locus for Golgi due to its movement on microtubules. We showed that the dynamic behavior of the fusome during the cell cycle, parallels Golgi dispersal and reassembly, and all these facts provide further strong support for the Golgi-association of the EMA granule and fusome.

(2) Figure 1F: is this image compressed?

We have now substituted the image in Figure 1F with a better image and have avoided the compression of the image.

(3) In the figure legends, are the sample sizes individual animals or individual sections? Please ensure that all figure legends for each figure panel consistently contain the sample size.

We have now included the number of measurements (N) in every figure legend. Each experiment was performed using samples from at least three different animals, and in most cases from more than three. This information has also been added to the Methods section under Statistics. In addition, N values are now consistently provided for each graph throughout the figures.

(4) Figure 2b/c: seemly likely based on the snapshot of different stages of cytokinesis that the "newly formed" visham is accurate, but without live imaging, this claim of "newly formed" is putative/speculative. It is OK if it is labeled as "putative" in the figure panel.

The behavior of the Drosophila fusome during mitosis was deduced without live imaging (deCuevas et al. 1998). We clarified that the conversion of a single mouse germ cell with one round fusome to an interconnected pair of cells with two round fusomes of greater total volume following mitosis is the basis for deducing that new fusome formation occurs each cell cycle. However, we agree with the reviewer that the phrase "newly formed" in the original label on Figure 2c suggested a specific mechanism of fusome increase that was not intended and this phrase has been removed entirely.

(5) Figure 2e/e is extremely difficult to follow. In order to improve the readability of these figure panels, can individual panels with a single stain be shown? The 'gap' between YFP+ sister cells is not immediately obvious in panel e or e" with the current layout. Since this is a key aspect of the author's claim about cleavage of the cyst, it would be best to make this claim more robust by showing more convincing images. In Figure 2E, the staining pattern of EMA needs to be clarified and described more fully in the text.

We mapped discontinuities in the microtubule connections, not the fusome or YFP. YFP is the lineage marker indicating that the cells of a single cyst are being studied. Consequently, no gap between YFP cytoplasmic expression is expected because only in the last example (figure E”), has fragmentation already occurred (and here there is a YFP gap). The acetylated tubulin gap proceeds fragmentation. The mitotic spindle remnants labeled by AcTub link the cells into two groups separated by a gap, which is clearly shown in the data images and in the third column where only the relevant AcTub from the cyst itself is shown. In response to the reviewers question about the fusome, which is not directly relevant to fragmentation, we have now provided images of the separate fusome channel and corresponding measurements for all three Figure 2E-E'' cysts in the supplementary Figure S4H. We have improved the text regarding this important figure to try and make it easier to follow, and also added a new example of a 10-cell cyst also in S2H (lower panels). We also added, movies allowing full 3D study of one of the 8 cell cysts and the new 10-cell cyst. I also suggest that the reviewer examine how the deduced mechanism of fragmentation explains previously published but not fully understood data on cyst fragmentation going back to 1998 as described in the expanded Discussion on this topic.

(6) It would be best to support the proposed model in Figure 2G (4+4+4) with microscopy images of a 12-cell or 16-cell cyst? Would these 12-cell or 16-cell cysts be too large to technically recover in a section?

Unfortunately the reviewer 's suggestion that 12- or 16-cell cysts are too large to recover and present convincingly is correct. Because our analysis depends on capturing lineage-labeled cysts specifically at telophase with acetylated-tubulin connections, the likelihood of obtaining the correct stage is very low. In addition, the dense packing of germ cells in the mouse gonad further limits our ability to fully reconstruct all the cells in large cysts, with difficulty increasing as cyst size grows.

However, as noted, we added a well-resolved 10-cell cyst—the largest size we could confidently analyze—in a 3D video in Supplementary Figure S2H (lower panel), which shows a 6 + 4 breakage pattern.

(7) We did not find a reference in the text for Figure 2G.

We have now provided reference for 2G in the text and in the discussion section.

(8) Line 189: ERAD is used as an acronym, but is not defined until the discussion.

We have now provided full form of acronym at its first usage in the text.

(9) Fig 3i/i': the increase of UPR pathway components, increasing expression during zygotene, is interesting to note, but is not commented enough in the text of the paper.

We have discussed this issue in the discussion section with specific reference to figure 3I. Please find the detailed discussion under the heading “Germ cell rejuvenation is highly active during cyst formation.”

(10) Please quantify DNMT3A expression levels in WT control vs Dazl KO germ cells in Figure 4a.

We have now quantified DNMT3A expression levels in WT control vs Dazl KO germ cells and have added the data in the Figure 4A.

(11) Please introduce the rationale behind selecting DazL KO for studying cyst formation (text in line 197). This comes out of nowhere.

True. We significantly expanded our discussion of Dazl and citations of previous work, including evidence that it can affect cyst structures like ring canals, in the Introduction.

(12) It would be best to stain WT control vs DazL KO oogonia in Figure 4a with 5mC antibodies to support their claim that DNA methylation might be affected in the mutants.

We respectfully disagree that this additional experiment is necessary within the scope of the current study. At the developmental stage examined (E12.5), germ cells in the Dazl mutant are clearly in an arrested and hypomethylated state, as supported by previous evidence (Haston et al. 2009).This initial experiments was designed to show that in our hands Dazl mutants show this known pkuripotency delay. However, the effects of Dazl mutation on female germline cyst development as it relates to polarity or the fusome was not studied before, and that is what the paper addresses, building on previous work.

Because our study does not focus on germ-cell epigenetic modifications but rather on the consequences of Dazl loss on germ cell cyst development, adding 5mC immunostaining would not substantially advance the main conclusions. The existing data and previous published work already provide sufficient background.

(13) Figure 4c: a very interesting figure, it would be best to quantify developmental pseudotime (perhaps using monocle3 analysis) and compare more rigorously the developmental stage of WT control vs DazL KO.

Developmental pseudotime, such as through Monocle3 analysis, might sometimes be valuable but involves assumptions that when possible are better addressed by direct experimental examination. Our conclusions regarding cyst developmental stage are supported by straightforward evidence rather to which computational trajectory inference would add little. Specifically, we have performed analysis of germ-cell methylation state, ring canal formation, pluripotency markers, UPR pathway activity assay (Xbp1 and Proteomic assay), Golgi-stress analysis and Pard3 which collectively document the developmental status of the WT and Dazl KO germ cells. These empirical data demonstrate the same developmental pattern reflected in Figure 4c, making the less reliable pseudotime-based computational method superfluous.

(14) Figure 4d has two panels labeled as "d".

We have now corrected the labelling of the figure

(15) Color coding in 4d, d', d" is confusing; please harmonize some visual presentation here.

We have now harmonized the visual representation of all the graph in figure 4

(16) Fig 4e' is labeled as DazL +/- but is this really a typo?

Thank you for pointing it out. We have now corrected the typo

(17) Figure F': typo labeled as E3.5, which is E13.5?

Thank you for pointing it out. We have now corrected the typo

(18) Figure F': was DazL KO mutant but no WT control.

The WT control was not provided to avoid the redundancy. Please refer to earlier figure 3A-B, Fig S3C and D and videos S3A and S3b to refer to WT control at every stage.

(19) Figure G: unusual choice in punctuation marks for cartoon schematic. No key to guide the reader for color-coded structures would be helpful to have something similar to 4h.

We have now provided the key to guide the readers in the mentioned figure 4G.

(20) The authors use WGA and EMA as interchangeable markers (Figure 5a) without fully explaining why they have switched markers.

Because it is germ cell specific, we used EMA as a fusome marker during the time when it is found up through E13.5. After that point we used WGA which is still usable, but also labels somatic cells. This rationale is explicitly described at the end of the section “Fusome is highly enriched in Golgi and vesicles”, where we state:

“EMA staining disappears from germ cells at E14.5 (Figure 1I). However, very similar (but non–germ-cell-specific) staining continued with wheat germ agglutinin (WGA) at later stages (Figure 1G, G’; Figure S1G).”

To ensure this is fully clear to readers, we have now added an additional statement in the start of the text section discussing the figure 5:

“For the reasons explained previously (see text for Figure 1G), WGA was used as a fusome marker beyond stage E14.5.”

(21) Figure 5b' is compressed.

We have now decompressed the image

(22) Line 267, Balbiani body is misspelled.

We have now corrected the spelling.

(23) The explanation of why the authors switch focus from DazL KO to DazL +/- is not adequately described. The authors should also explain the phenotype of the DazL +/- animals or reference a paper citing the hets are sterile or subfertile.

We have now added the explanation of why Dazl KO is used in our introduction section where we have mentioned the phenotype of Dazl homozygous and heterozygous mouse.

(24) Is Figure 5i actually DazL +/-? It is not labeled clearly in the text, the figure legend, or the figure panel.

We have now labelled the figure correctly in figure and in the legend.

(25) The paper ends abruptly at line 275 with no context or summary.

The manuscript does not end at line 275; the apparent interruption is due to a page break occurring immediately before the beginning of the Discussion section. We hope that continuation is fully visible in the reviewer 1 (your) version of the PDF.

**Reviewer #2 (Recommendations for the authors):**
(1) Line 93: Fig. 1B: DDX4 marks germ cells; do all the red and yellow cells in the NE inset originate from the same PGC? There are only 2 cells marked in yellow among the group of red cells. Is it a z-projection issue? Or do they come from different PGCs?

This experiment used vasa staining to identify all germ cells, which are produced by multiple PGCs. Green labeling is a lineage marker derived from a single PGC (due to the low frequency of tamoxifen-activated labeling). Consequently, the two yellow cells observed in the NE inset of Fig. 1B represent YFP-labeled germ cells (YFP + DDX4 double-positive) that have arisen from a single, lineage-traced PGC. This approach, introduced in 2013, is described in the Methods, and represents the field's single largest technical advance that has made it possible to analyze mouse germ cell development at single cell resolution.

To ensure clarity, we have added a brief explanatory note to the figure legend indicating that yellow cells represent the lineage-traced progeny of a single PGC, while the red staining marks all germ cells.

(2) Line 96: Figure 1C vs 1C'. The difference between female and male Visham is not obvious, although quantification shows a clear difference. How was the quantification made? Manual or automatic thresholding? Would it be possible to show only the Visham channel?

We thank the reviewer for pointing out this problem. We have now more clearly described in the text that the female fusome increases in some cells with close attachments to other cells (future oocytes) and decreases in distant nurse cells. It branches due to rosette formation.. In males, the fusome remains much like the initial EMA granules present in early germ cells, with only fine and difficult to see connections. The quantification shown in Figures 1C and 1C′ was performed manually, based on the presence of either (i) fused, branched EMA-positive fusome structures or (ii) dispersed, punctate EMA granules. This assessment was carried out across multiple E13.5 male and female gonad samples to ensure robustness. To facilitate independent evaluation, we have already provided supplementary videos S3B1 and S3B2, which display the EMA-stained E13.5 male and female gonads in three dimensions. These videos allow the structural differences to be examined more clearly than in static images.

In response to the reviewer’s request, we now additionally include the single-channel fusome image in Supplementary Figure S1E′. This presentation highlights the fusome signal alone and further clarifies the morphological differences underlying the quantification.

(3) L118: Figure 2A, third row = 2-cell cyst? Please specify PCNT in the legend.

We appreciate the reviewer’s observation. In Figure 2A (third row), the cells were not specifically labeled as a 2-cell cyst; rather, the intention was to illustrate the presence of two distinct centrosomes positioned on a fused fusome structure, a configuration we frequently observe.

We have now updated the figure legend to explicitly define PCNT.

(4) L169: Missing reference to S3B and video S3B1?

We have now included the reference to S3B1 and S3B2 in the text and in the legend

(5) L170: Please describe the graph in the Figure 3D legend.

We have now described the Graph in the legend

(6) L171: Would it be possible to have a close-up showing both Pard3 and Visham in a ringlike pattern related to RACGAP (RC) staining? The images are too small.

It is difficult to capture this relationship perfectly in a two dimensional picture. The images represent the maximum close-up possible that still includes enough relevant area for the necessary conclusions. We have now provided additional three close-up images exclusively for ring-canal and Pard3 association in the supplementary Figure S3C for further clarity. However, we also note that the quality of the image permits the reader of a pdf to zoom and to visualize the images in great detail.

(7) L181: Wrong reference, should be 3 then 3I.

Thank you for pointing it out, we have now corrected the reference.

(8) L199: In Figure S4B, was DNMT3 staining quantified? Red intensity differs globally between images; use the somatic red level as a reference? Note: EMA seems higher in Dazl- vs. WT?

We have now performed quantification of DNMT3 staining, which is presented in Figure 4A. While the red intensity (DNMT3 or EMA) can appear to differ between images, this variation can result from biological differences between tissues or minor technical variability despite using consistent microscope settings. To account for this, we normalized the staining intensity using the somatic cell signal as an internal reference, ensuring that the quantification reflects genuine differences between WT and Dazl-/- samples rather than global intensity variation.

(9) L229: Should be "proteasome."

We have now corrected the spelling error.

(10) L233: Quantify fragmentation of Gs28? EMA doesn't seem affected. Could you quantify both Gs28 and EMA? Images are too small.

We thank the reviewer for this suggestion. While the current images are small, they can be examined in detail using zoom to visualize the structures clearly. As noted, EMA staining is not affected, (we agree) as cells are in arrested state. This arrested state creates stress on Golgi. The fragmentation of Gs28-labeled Golgi membranes is a classical indicator of Golgi stress, even though the fragmented membranes may remain functionally active. Our results show that Dazl deletion specifically affects Golgi in germ cells, while Golgi in neighboring somatic cells appears healthy. To quantify this effect, we have now included manual quantification of Golgi fragmentation in Figure 4F, assessing tissues for the presence of fragmented versus intact Golgi structures. This confirms that Golgi fragmentation is a germ cell–specific phenotype in Dazl– samples, while pre-formed EMA-positive fusomes remain unaffected but probably in arrested state.

(11) L237: Figure 4F graph shows E3.5, not E13.5.

We have now corrected the typo in the figure

(12) L257: Figure 5D: quantify as in 5A? overlap?

Yes, it's an overlap and shown as two separate image with ring canal for better clarity. We have now quantified the image and have produced combined graph for fusome and pard3 in Figure 5A graph.

(13) L261: Figure 5E-E': black arrowhead not mentioned in legend.

We have now mentioned the black arrowhead in the legend

(14) L262: Figure 5C: arrowhead not mentioned in legend. Figure 5F: oocyte appears separated from nurse cells compared to 5C.

Yes, that may happen as cysts undergo fragmentation; what matters is all cells are lineage labelled and hence are members of a single cyst derived from one PGC.

(15) L263: Figure 5G has no legend reference; nurse cells are not outlined as in 5C.

We have now outlined the nurse cells and have added the reference to the graph in the legend.

(16) L279: "The fusome and Visham and both..." should be replaced with "Both fusome and Visham...".

We have now replaced the term Visham with fusome as suggested by reviewers and editor. We updated the statement to correct the grammatical error.

(17) L1127: Video S3B1: It is unclear what to focus on.

We have now added the Rectangle area and arrow to highlight what to focus on

(18) L1128: Video "S3B1" should be "S3B2."

We have now corrected the legend

(19) Finally: curiosity question: have the authors tried to use known markers of the Drosophila fusome in mice, such as Spectrin or other markers described in Lighthouse, Buszczak and Spradling, Dev Bio, 2008? And conversely, do EMA and WGA label the fusome in Drosophila?

Yes, we and others used the most specific markers of the Drosophila fusome such alpha-spectrin, adducin-like Hts, tropomodulin, etc. to search for fusomes in vertebrate species. It was unsuccessful in clarifying the situation, because Hts and alpha-spectrin in Drosophila and other insects generate a protein skeleton that stabilizes the fusome and is easily stained. But this structure is simply not conserved in vertebrates. The polarity behavior of the fusome, it core developmental property, is conserved, however. The mammalian fusome still acquires and maintains cyst polarity, and goes even farther and reflects both initial cyst formation and cyst cleavage, before marking oocyte vs nurse cell development in the smaller cysts. Expression of the inner microtubule-rich portion of the fusome, its Par proteins, and many ER-related and lysosomal fusome proteins are mostly conserved but their ability to mark the fusome alone varies with time and context (only some of the examples are shown in Figure 3I'). Nearly all of the proteins identified in Lighthouse et al. 2008 are expressed. These proteins may be involved in rejuvenation as studied here. We modified the first section of the Discussion to explicitly compare mouse, Xenopus and Drosophila fusomes, which was not possible before this work.

**Reviewer #3 (Recommendations for the authors):**
The authors should either revise the conclusions or add additional evidence to support their claims. In addition, minor corrections are listed below.

We have added additional evidence as noted in responses above, and revised some claims that were stated inaccurately. In addition, we have attempted to clarify the evidence we do present, so that its full significance is more easily grasped by readers.

(1) Lines 20-21 are unclear - the cyst doesn't get sent into meiosis, each oocyte does.

Research is showing that it's more complicated than that. All cyst cells enter "pre-meiotic S phase", and most cell cycles are conventionally considered to start after the previous M phase-

i.e. in G1 or S, not in the next prophase, an ancient view limited just to meiosis. Absent this old tradition from meiosis cytology, pre-meiotic S would just be called meiotic S as some workers on meiosis do. In addition, in different species, nurse cells diverge from meiosis on different schedules, including many much later in the meiotic cycle. Two cyst cells in Drosophila fully enter meiosis by all criteria, the oocyte and one nurse cell that only exits in late zygotene. In Xenopus and mouse, scRNAseq shows that many cyst cells enter meiosis up to leptotene and zygotene, including nurse cells that specifically downregulate meiotic genes during this time, possibly to assist their nurse cell functions, while others remain in meiosis even longer (Davidian and Spradling, 2025; Niu and Spradling, 2022). Eventually, only the oocytes within each fragmented mouse cyst complete meiosis.

(2) Many places in the manuscript abbreviations are never defined or not defined the first time they are used (but the second or third time): Line 23-ER, Line 29-UPR, Line 33-PGC (not defined until line 45), Line 79-EGAD.

We have defined full acronyms now upon their first occurrence.

(3) Line 5 should be the pachytene substage of meiosis I.

We have now updated the statement to “In pachytene stage of meiosis I…”

(4) Line 59-61 - this statement needs a reference(s).

These statements are a continuation from the references cited in the previous statements. However, for further clarity we have again cited the relevant reference here (Niu and Spradling, 2022).

(5) Line 80 - should it be oocyte proteome quality control?

We have now updated the statement to “Oocyte proteome quality control begins early”.

(6) Line 87 - in this case, EMA does not stand for epithelial membrane antigen (AI will call it that, but it is not correct). I believe it originally was the abbrev for (Em)bryonic (a)ntigen, though some papers call it (e)mbryonic (m)ouse (a)ntigen. And the reference here is Hahnel and Eddy, 1986, but in the reference list is a different paper, 1987 (both refer to EMA-1).

We have now updated the acronym EMA-1 in corrected form and have corrected the citation.

(7) Line 176 - RNA seq.

We have now updated the statement to “We performed single cell RNA sequencing (scRNA seq) of mouse gonad”.

(8) Line 181 - Figure 4E and 4I should be 3E and 3I.

We have now updated the figure reference in the text to correct one.

(9) Line 183 - missing period.

Added.